# Buck-Boost/Flyback Hybrid Converter for Solar Power System Applications

**Sheng-Yu Tseng *** and **Jun-Hao Fan**

Department of Electrical Engineering, Chang Gung University, Guishan District, Taoyuan City 33302, Taiwan; m0921014@cgu.edu.tw
* Correspondence: sytseng@mail.cgu.edu.tw; Tel.: +886-3-211-8800 (ext. 5706)

**Abstract:** This paper proposes a hybrid converter to supply power from solar power source to load. Since power is generated by solar power, which depends on the intensity of solar power, the power generated by the solar power does not keep at a constant power. Therefore, the proposed system needs a battery to balance power between solar power and load. When the proposed one uses the battery to balance powers, the proposed circuit requires a charger and discharger. To simplify the proposed converter, a buck-boost converter and flyback converter can be combined to implement the battery charging and discharging functions. With this approach, the proposed converter can be operated with zero-voltage switching (ZVS) at turn-on transition to reduce switching loss of switch when the proposed one is operated in the discharging mode. In addition, the proposed hybrid converter has several merits, which are less component counts, lighter weight, smaller size and higher conversion efficiency. As compared with the conventional counterparts with hard-switching circuit, the proposed one can increase conversion efficiency of 4% and achieve efficiency of 85% under full load condition when the proposed one is operated in the discharging mode. Experimental results which are obtained from a prototype with output voltage of 10 V and maximum output power 20 W have been implemented to verify its feasibility. It is suitable for an electronic sign indicating LED within 200 W, which is used in the night time.

**Keywords:** charger; discharger; ZVS; buck-boost converter; flyback converter; solar power and battery

## 1. Introduction

Nowadays, due to a drastic increase in the demand for electricity, it leads to rapid and depletion of fossil fuels. In particular, when Taiwan Semiconductor Manufacturing (TSMC) and Google data center were built in Taiwan, electricity demand was increased by 1/3 within 5 years. As a result, the power processor adopts renewable energy sources as its input sources. In the renewable energy sources, such as solar power, wind turbine and fuel cell, they have been widely applied to a switching-mode converter for generating electric power to load [1,2]. They include power generation for grid connection, electric vehicles, water pumps, battery charger and discharger, traffic signals, street-lighting, electronic signs, and so on.

Due to the most recent development of light emitting diodes (LEDs) technology, it possesses many advantages, such as smaller size, longer lifetime, lower maintenance costs and greater strength against breakage [3–6]. Therefore, LEDs have widely used in our daily lives. They are suitable for indoor and outdoor energy-saving lighting applications, such as automotive taillights, thin film transistor liquid crystal display (TFT-LCD) backlight, traffic signals, streetlights, electronic signs [7–10]. In particular, the electronic sign or streetlight is used in the night time. It is suitable for solar power sources to supply power when solar power sources are adopted in the electronic sign or streetlight system. That is, the power system uses a charger to store energy from solar power sources to battery in the daytime, while the one adopts a discharger to release energy from battery to LEDs in the

night time. The proposed power system simultaneously needs a charger and discharger, as shown in Figure 1.

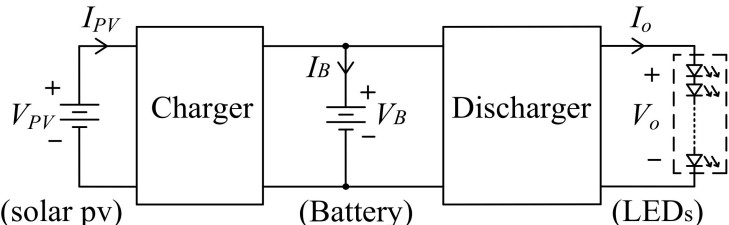

**Figure 1.** Block diagram of solar power system for electronic signor street-lighting application.

Since the proposed power system adopts PV sources as its input source, the one needs the charger and discharger for electronic sign or streetlight application. The output voltage of solar power source is less than or greater than that of battery (8 V~12 V). Therefore, it simultaneously needs a step-up and step-down converter, such as buck-boost, 'cuk, zeta and sepic converter [11–13]. As compared with 'cuk, zeta and sepic converter, buck-boost converter possesses a simpler circuit topology. It is chosen as the charger for battery system, as shown in Figure 2. Since the flyback converter possesses many merits, which are a simpler circuit topology, wider ranges of voltage ratio between input voltage and output voltage and a lower cost, it can be applied to the solar power system or battery system. Therefore, the proposed power system can adopt buck-boost converter and flyback converter to implement battery charging and discharging functions, simultaneously, as shown in Figure 3.

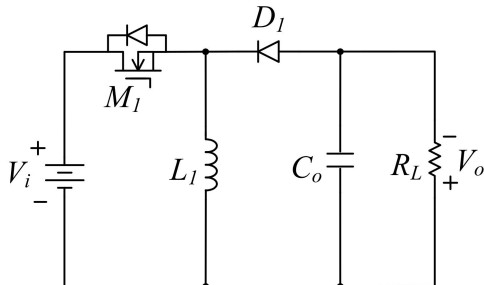

**Figure 2.** Schematic diagram of buck-boost converter for battery charging applications.

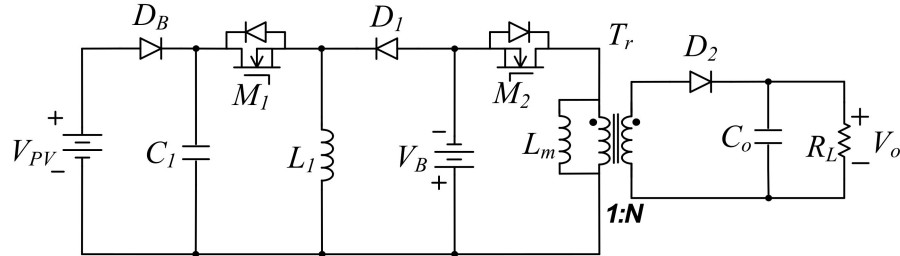

**Figure 3.** Schematic diagram of buck-boost/flyback hybrid converter for battery charging and discharging applications.

When the proposed power system uses the flyback converter as the discharger, the leakage inductor of transformer will induce a spike voltage across switch. As a result, it will generate an extra switching loss. To avoid this problem, an active clamp circuit can be added into the flyback converter to recover leakage inductor energy [14–17], as shown in Figure 4. In order to simply circuit topology, switches of the charger and discharger can be respectively integrated, as shown in Figure 5. From Figure 5, the proposed hy-

brid converter can use a less component count to implement the battery charging and discharging functions, simultaneously. Therefore, the proposed power system can reduce cost, weight and size. Furthermore, the proposed one can be operated in ZVS at turn-on transition to increase conversion efficiency. It is suitable for a PV power system.

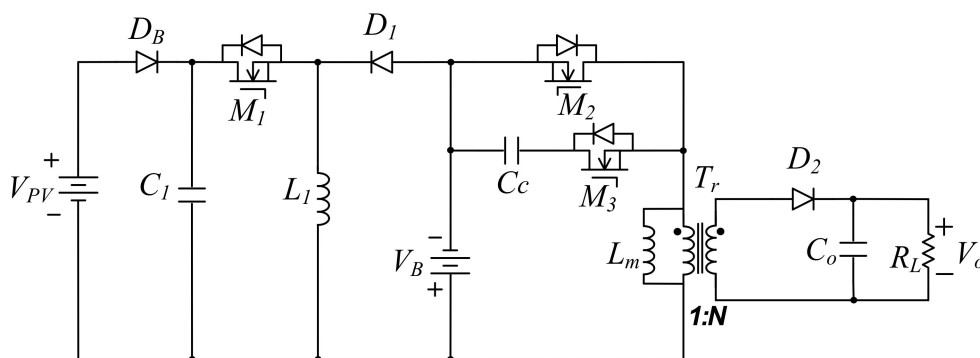

**Figure 4.** Schematic diagram of buck-boost/flyback converter with active clamp circuit for solar power applications.

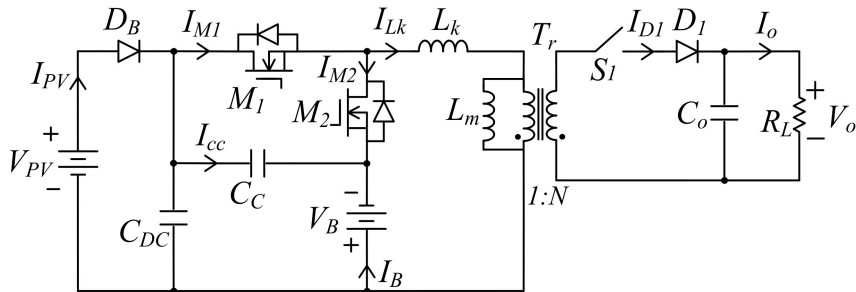

**Figure 5.** Schematic diagram of the proposed hybrid converter for solar power applications.

## 2. Derivation of the Proposed Hybrid Converter

The proposed hybrid converter adopts a buck-boost converter as the battery charger and a flyback converter as the battery discharger, as shown in Figure 4. In order to simplify the proposed power system, a bidirectional buck-boost converter is used and an active clamp circuit is introduced into flyback converter to increase conversion efficiency, as shown in Figure 6. From Figure 6, since the battery charger and discharger in the proposed power system are operated in complementary, switch $S_1$ is added into the proposed one to keep the battery charging and discharging functions. When switch $S_1$ is regarded as the operational mode switch, switches $M_2$ and $M_3$ shown in Figure 6 can be merged as a switch $M_2$ illustrated in Figure 5, while switches $M_1$ and $M_4$ can be merged as a switch $M_1$. The inductor $L_1$ can be also merged with the magnetizing inductor $L_m$ of transformer. With this approach, the proposed power system can use a less component count to achieve the battery charging and discharging functions.

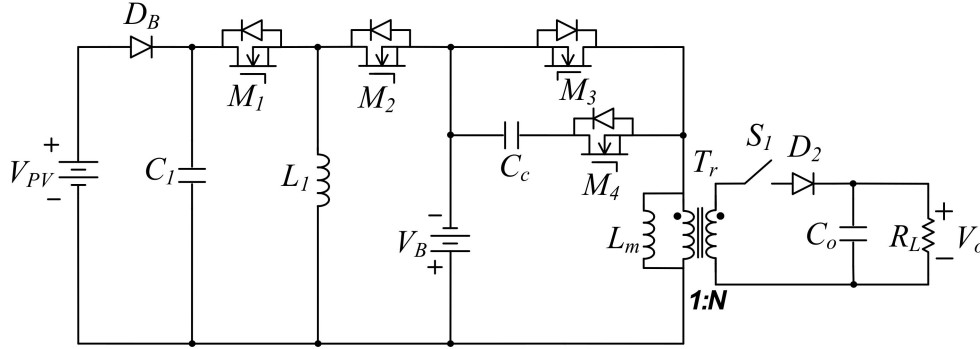

**Figure 6.** Schematic diagram of the convention bidirectional buck-boost/flyback hybrid converter for solar power applications.

When the proposed power system is operated in the charging mode, switch $S_1$ is turned off. The equivalent circuit of the proposed one is the same as a buck-boost converter, as shown in Figure 7a. In Figure 7a, the magnetizing inductor of transformer $T_r$ is respected as the inductor of the buck-boost one, where topology of the proposed hybrid converter operated in the charging mode is highlighted with thick line. When the proposed one is operated in discharging mode, switch $S_1$ is turned on, its equivalent circuit is the same as active clamp flyback converter, as shown in Figure 7b. In Figure 7b, the proposed hybrid one is highlighted with thick line. Therefore, the proposed power system can be respectively operated in different modes by the operational conditions of switch $S_1$.

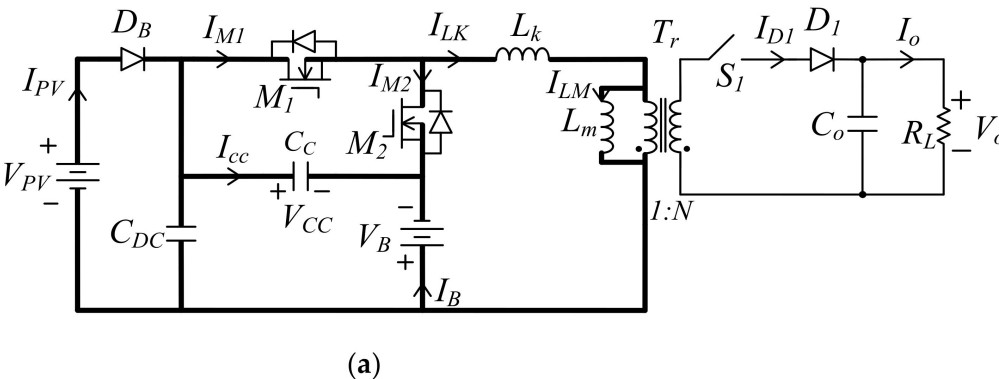

(**a**)

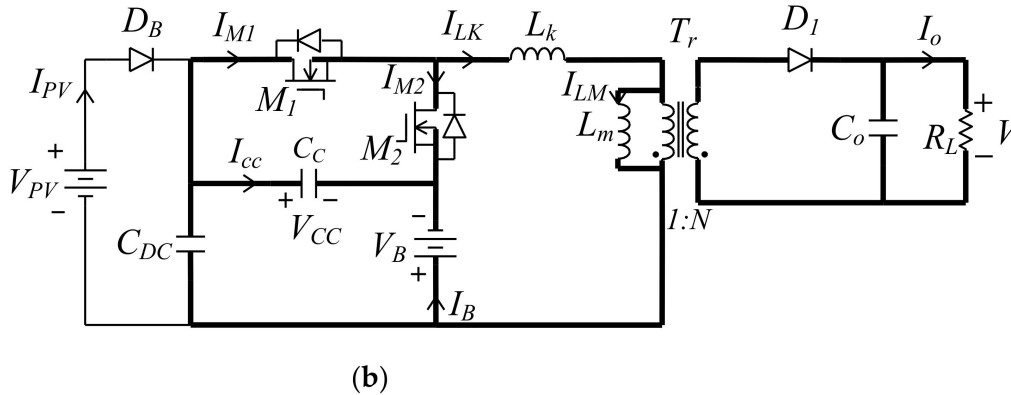

(**b**)

**Figure 7.** Schematic diagram of the proposed hybrid converter operated in (**a**) the charging mode, and (**b**) the discharging mode.

## 3. Operational Principle of the Proposed Power System

The proposed hybrid converter can be divided into two operational modes: the charging mode and the discharging mode. Its equivalent circuit of the different operational modes is shown in Figure 7. Since the proposed hybrid converter can be operated in two different modes, its operational principle for each operational mode is also described in the following, respectively.

The charging mode

When the proposed hybrid converter is operated in the charging mode, its operational modes can be divided into 6 modes. The equivalent circuit of each operational mode and conceptual waveforms are plotted in Figures 8 and 9, respectively. The power flow of each operational mode in the proposed hybrid converter is highlighted with thick line, as shown in Figure 9. In the following, each operational mode is briefly described.

Mode 1 [Figure 9a: $t_0 \leq t < t_1$]: Before $t_0$, switch $M_1$ is in the turn-on state and switch $M_2$ is in the turn-off state. During this time interval, switch current $I_{M1}$ abruptly increases from 0A to the initial value of inductor $L_m$ operated in continuous condition mode (CCM). When $t = t_0$, switch $M_1$ is still in the turn-on state, and $M_2$ is kept in the turn-off state. Since inductance $L_m$ is further greater than $L_k$, voltage $V_{PV}$ is approximately applied to inductance $L_m$. During this time interval, inductor $L_m$ is in the stored energy state. Inductor current $I_{Lm}$ linearly increases. The charging current $I_B$ equals 0A.

Mode 2 [Figure 9b: $t_1 \leq t < t_2$]: At $t = t_1$, switch $M_1$ is turned off and switch $M_2$ is still in the turn-off state. During this time interval, capacitor voltage $V_{CM2}$ is discharging from $(V_{PV} + V_B)$ to 0V, while $V_{CM1}$ is charged from 0V to $(V_{PV} + V_B)$. Within this mode, switch current $I_{M1}$ abruptly decreases from the maximum value to 0A, and current $I_{M2}$ quickly increases from 0V to the maximum value.

Mode 3 [Figure 9c: $t_2 \leq t < t_3$]: At $t = t_2$, body diode $D_{M2}$ is in the forwardly bias state. Energy stored in magnetizing inductor $L_m$ is released through battery and body diode $D_{M2}$. Inductor current $I_{Lm}$ linearly reduces. The battery is in the charging state.

Mode 4 [Figure 9d: $t_3 \leq t < t_4$]: At $t_3$, switch $M_2$ is turned on and $M_1$ is kept in the turn-off state. Since body diode $D_{M2}$ is forwardly biased before $t = t_3$, switch $M_2$ is operated at ZVS at turn-on transition. Energy stored in magnetizing inductor $L_m$ is still in the released energy state. Current $I_{Lm}$ is equal to charging current $I_B$, and its value linearly increases.

Mode 5 [Figure 9e: $t_4 \leq t < t_5$]: At $t = t_4$, switch $M_2$ is turned off, and $M_1$ is kept at the turn-off state. Since energy stored in inductor current $I_{Lm}$ is released through battery and body diode $D_{M2}$, switch voltage $V_{M1}$ is kept at 0V. Capacitor voltage $V_{CM1}$ is also kept at $(V_{PV} + V_B)$. Inductor current $I_{Lm}$ linearly reduces.

Mode 6 [Figure 9f: $t_5 \leq t < t_6$]: At $t = t_5$, switch $M_1$ is turned on, and $M_2$ is kept in the turn-off state. Due to body diode $D_{M2}$ is in the forwardly bias state before $t = t_5$, switch voltage $V_{M1}$ abruptly varies from $(V_{PV} + V_B)$ to 0V, switch voltage $V_{M2}$ fast changes from 0V to $(V_{PV} + V_B)$. Switch current $I_{M1}$ suddenly increases to the initial value when the proposed converter is operated in CCM. Switch current $I_{M2}$ and battery current $I_B$ simultaneously decreases to 0A. Since this time interval is very short, current $I_{M1}$ is kept at the initial value and current $I_{M2}$ and battery current $I_B$ are sustained at 0A. When $t = t_6$, a new switching cycle will start.

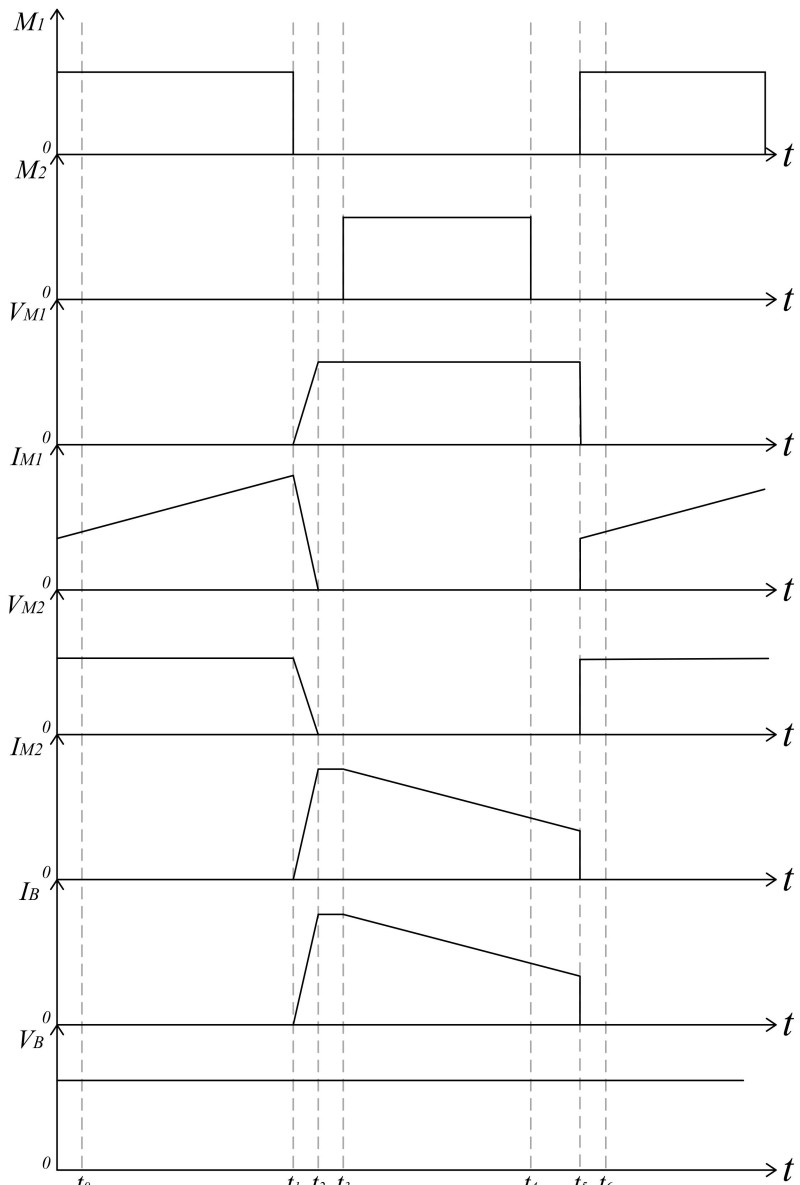

**Figure 8.** Conceptual waveforms of the propose hybrid converter operated in the charging mode.

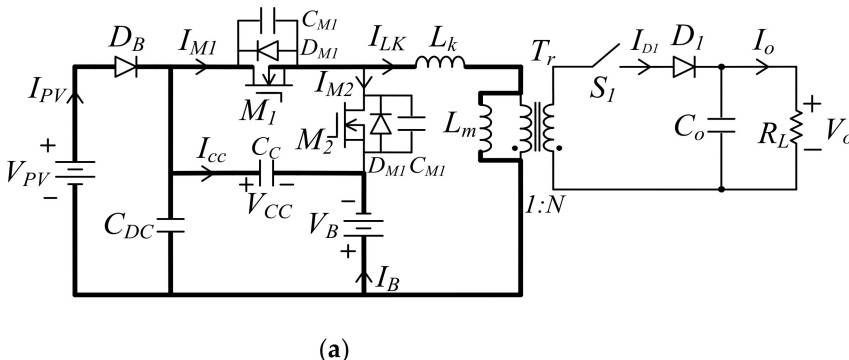

(**a**)

**Figure 9.** *Cont.*

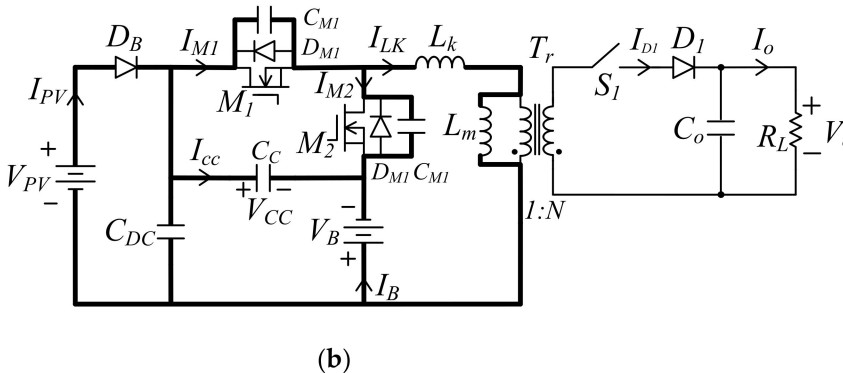

(**b**)

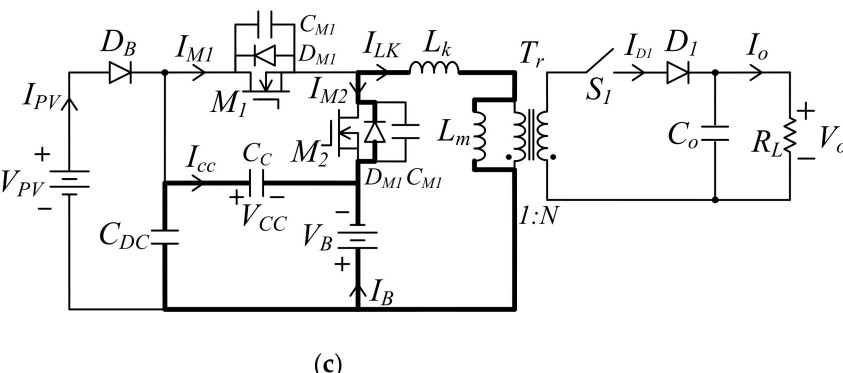

(**c**)

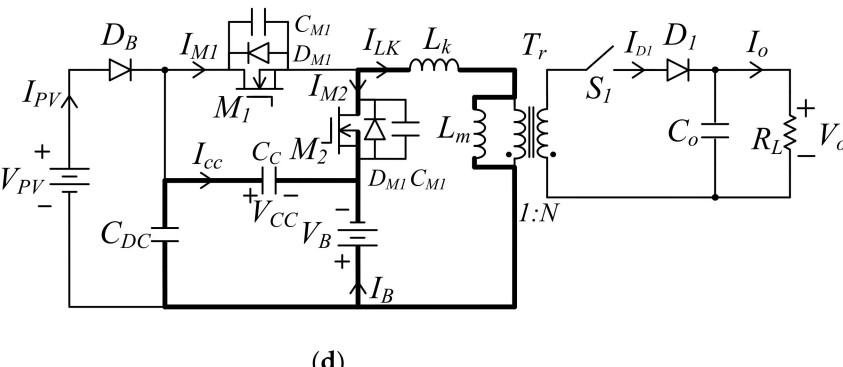

(**d**)

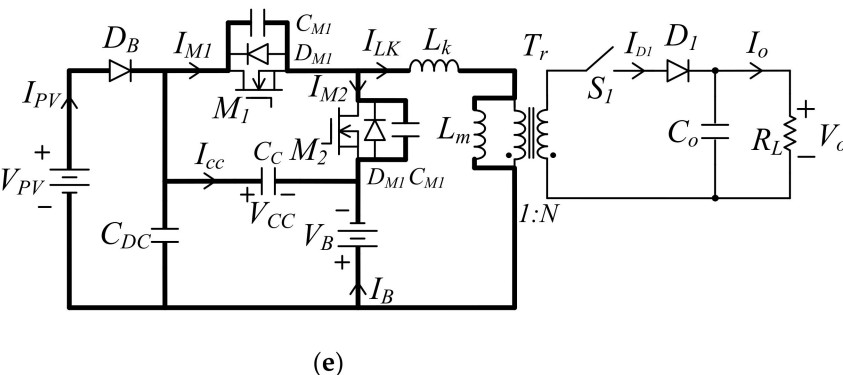

(**e**)

**Figure 9.** *Cont.*

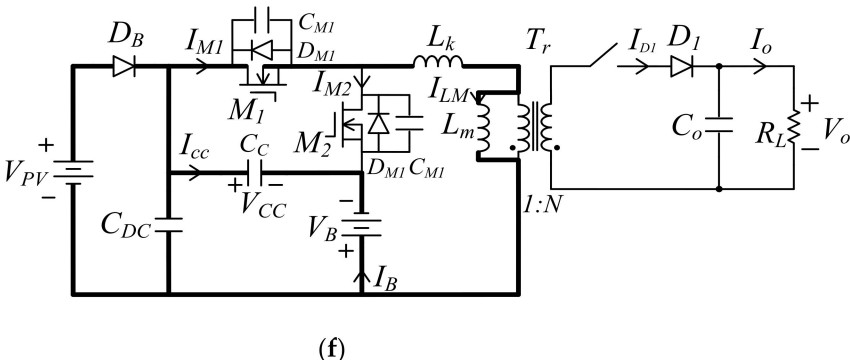

(f)

**Figure 9.** Equivalent circuit of the proposed hybrid converter operated in the charging mode over one switching cycle; (**a**) Mode 1 ($t_0 \leq t < t_1$); (**b**) Mode 2 ($t_1 \leq t < t_2$); (**c**) Mode 3 ($t_2 \leq t < t_3$); (**d**) Mode 4 ($t_3 \leq t < t_4$); (**e**) Mode 5 ($t_4 \leq t < t_5$); (**f**) Mode 6 ($t_5 \leq t < t_6$).

The discharging mode

When the proposed hybrid converter is operated in the discharging mode during the night time, PV arrays does not generate power to supply battery. Furthermore, the proposed one with battery is required to supply energy to lighting system. According to the previously requirements, switch $S_1$ is turned on and PV arrays are not to supply power to load by diode $D_B$. Its equivalent circuit is implemented by flyback with active clamp circuit, as shown in Figure 7b. When the proposed hybrid converter is formed with the active clamp flyback converter, its operational mode is divided into eight modes. Its key component waveform is illustrated in Figure 10. In addition, equivalent circuit of each operational mode is depicted in Figure 11. In the following, each operational mode is described briefly.

Mode 1 [Figure 11a: $t_0 \leq t < t_1$]: Before $t_0$, switch $M_2$ is in the turn-on state and switch $M_1$ is in the turn-off state. Switch current $I_{M2}$ fast varies from 0A to the initial value of inductor $L_m$ in the proposed converter operated in CCM. When $t = t_0$, switch $M_2$ is still in the turn-on state, and $M_1$ is kept in the turn-off state. Switch current $I_{M2}$ is equal to the initial value of inductor current $I_{Lm}$. During this time interval, magnetizing inductor $L_m$ is in the stored energy state. Inductor current $I_{Lm}$ linearly increases. Since diode $D_1$ is reversely biased, load power is supplied by output capacitor $C_o$.

Mode 2 [Figure 11b: $t_1 \leq t < t_2$]: At $t_1$, switch $M_2$ is turned off and switch $M_1$ is still in the turn-off state. Since inductor current $I_{LK}$ must be operated in the continuous condition through capacitor $C_{M1}$ and $C_{DC}$, Capacitor $C_{M1}$ is discharged and its voltage $V_{M1}$ varies from ($V_o/N + V_B$) to 0V. Within this mode, capacitor $C_o$ supplies power to load. Switch current $I_{M2}$ abruptly decreases to 0A, while current $I_{M1}$ suddenly reduced to the negative maximum value. Capacitor current $I_{CC}$ also varies from 0V to its maximum value.

Mode 3 [Figure 11c: $t_2 \leq t < t_3$]: When $t = t_2$, body diode $D_{M1}$ is forwardly biased and diode $D_1$ is also in the forwardly bias. Inductor voltage $V_{Lm}$ is clamped at ($-V_o/N$). During this time interval, leakage inductor $L_k$ and capacitor $C_C$ form a resonant network. Inductor current $I_{Lk}$ varies with the resonant form from the maximum value to the negative maximum value. Energy stored in the magnetizing inductor $L_m$ is released through secondary winding of transformer $T_r$ and diode $D_1$ to load.

Mode 4 [Figure 11d: $t_3 \leq t < t_4$]: At $t = t_3$, switch $M_1$ is turned on and $M_2$ is kept in the turn-off state. Because body diode $D_{M1}$ is in the forwardly bias state before $t = t_3$, switch $M_1$ is operated with ZVS at turn-on transition. Within this mode, inductor $L_k$ and capacitor $C_c$ connects in series to generate the resonance. Inductor current $I_{Lk}$ is still in the resonant state, and current $I_{Lm}$ linearly reduce to release the energy stored in the magnetizing inductor $L_m$.

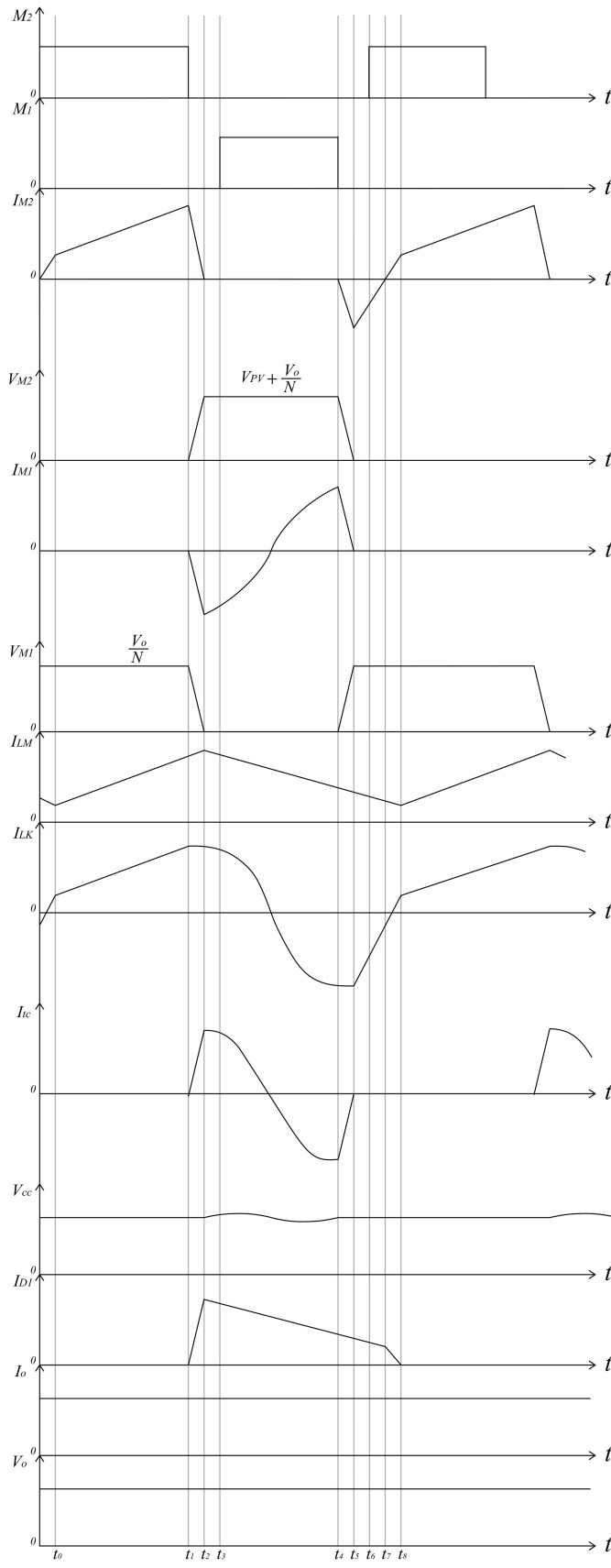

**Figure 10.** Conceptual waveforms of the propose hybrid converter operated in the discharging mode.

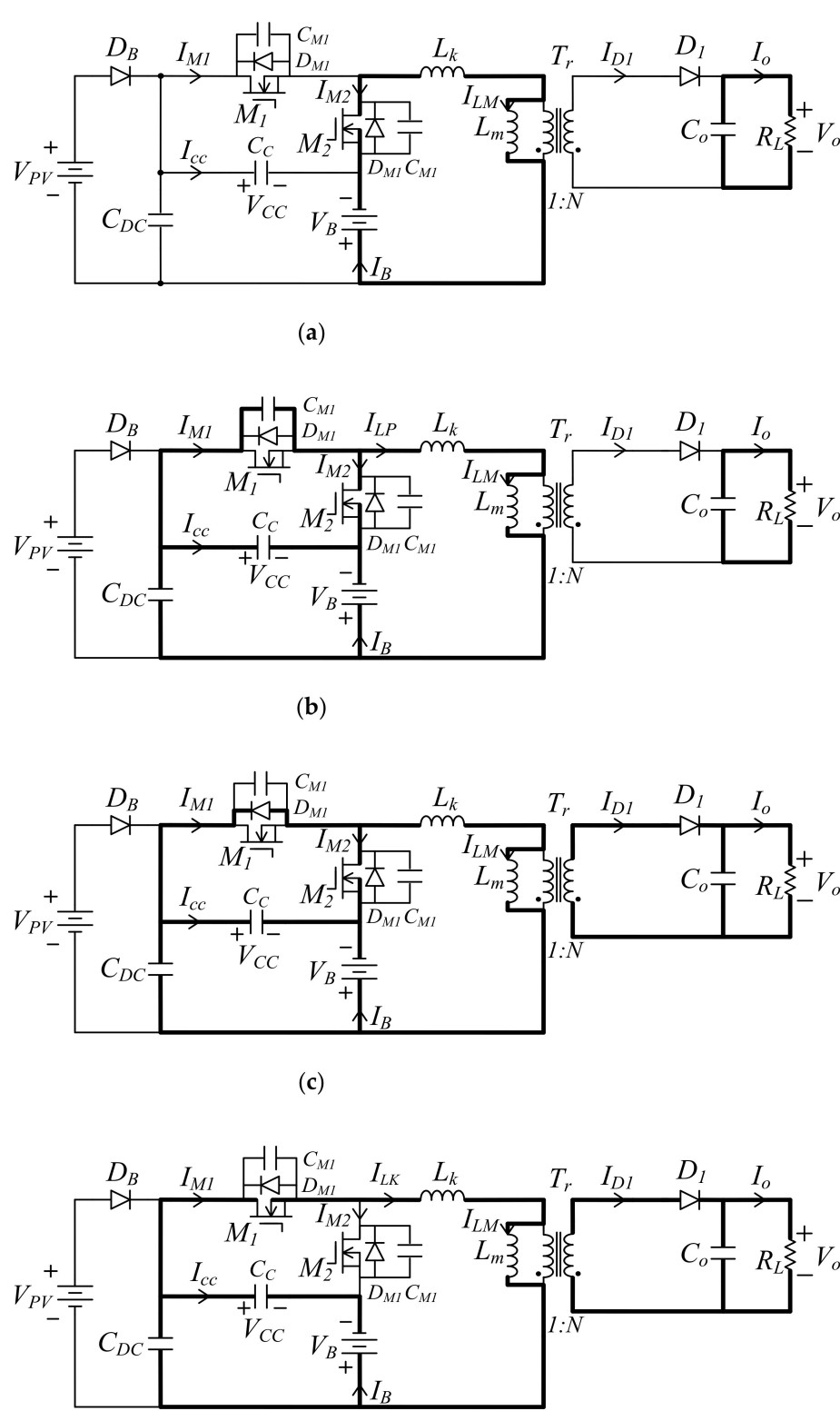

(**a**)

(**b**)

(**c**)

(**d**)

**Figure 11.** *Cont.*

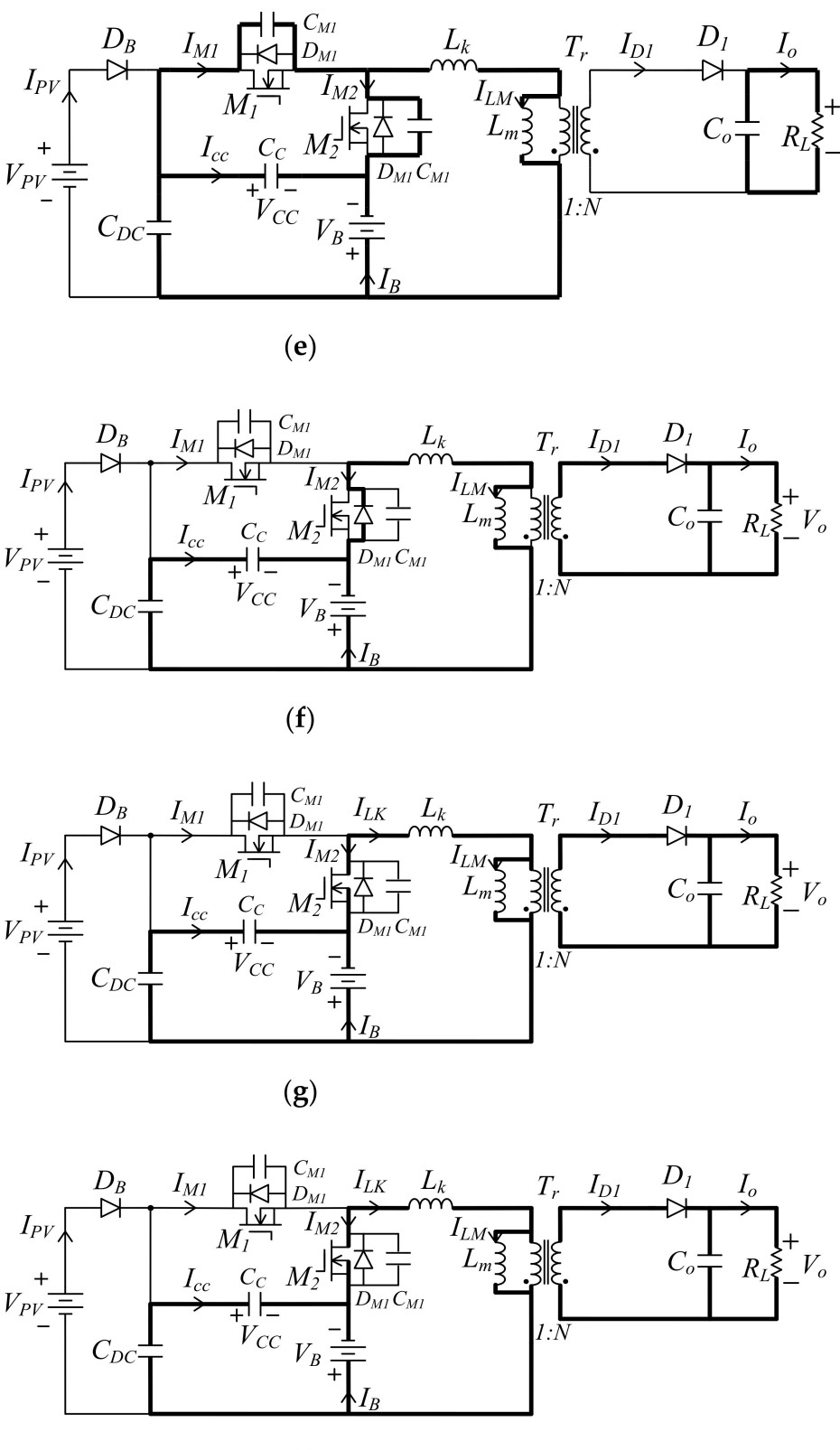

**Figure 11.** Equivalent circuit of the proposed hybrid converter operated in the discharging mode over one switching cycle. (**a**) Mode 1 ($t_0 \leq t < t_1$); (**b**) Mode 2 ($t_1 \leq t < t_2$); (**c**) Mode 3 ($t_2 \leq t < t_3$); (**d**) Mode 4 ($t_3 \leq t < t_4$); (**e**) Mode 5 ($t_4 \leq t < t_5$); (**f**) Mode 6 ($t_5 \leq t < t_6$); (**g**) Mode 7 ($t_6 \leq t < t_7$); (**h**) Mode 8 ($t_7 \leq t < t_8$).

Mode 5 [Figure 11e: $t_4 \leq t < t_5$]: At $t_4$, switch $M_1$ is turned off, and $M_2$ is kept at the turn-off state. Since capacitor $C_{M1}$ enters the charging state, capacitor voltage $V_{M1}$ varies from 0V to $(V_o/N + V_B)$. Moreover, capacitor voltage $V_{M2}$ works in the discharging state, voltage $V_{M2}$ change from $(V_o/N + V_B)$ to 0V. Within this mode, energy stored in the magnetizing inductor $L_m$ releases through diode $D_1$ to load. Inductor current $I_{Lm}$ linearly reduces.

Mode 6 [Figure 11f: $t_5 \leq t < t_6$]: When $t = t_5$, capacitor voltage $V_{M1}$ is clamped at $(V_o/N + V_B)$, while voltage $V_{M2}$ is kept at 0V. At the moment, body diode $D_{M2}$ is forwardly biased. During this time interval, inductor current $I_{Lk}$ equals to current $I_{M2}$. Their values abruptly varies from the negative maximum value to 0V. The magnetizing inductor $L_m$ is still in the discharging energy state, and its current $I_{Lm}$ linearly decreases.

Mode 7 [Figure 11g: $t_6 \leq t < t_7$]: When $t = t_6$, switch $M_2$ is turned on, and switch $M_1$ is still kept in the turn-off state. Since body diode $D_{M2}$ is in the forwardly bias state before $t_6$, switch $M_2$ is operated with ZVS at turn-on transition. During this time interval, current $I_{LK}$ (=$I_{M2}$) changes from a negative value to 0V. The magnetizing inductor $L_m$ is still in the released energy state through diode $D_1$ to load. Its current $I_{Lm}$ linearly reduces.

Mode 8 [Figure 11h: $t_7 \leq t < t_8$]: At $t_7$, switch $M_2$ is in the turn-on state and $M_1$ is in the turn-off state. During this time interval, inductor current $I_{LK}$ varies from 0A to the initial value. The magnetizing inductor $L_m$ is kept in the released energy state. Therefore, current $I_{Lm}$ linearly decreases. When operational mode is at the end of mode 8, one new switching cycle will start.

## 4. Design of the Proposed Hybrid Converter

The proposed hybrid converter includes a charger and discharger. When the proposed one is operated as the charger, its equivalent circuit is the same as buck-boost converter. Moreover, its equivalent circuit is formed with an active clamp flyback converter for the discharger. Since the proposed one is composed with charger and discharger, its design must satisfy requirements of each converter. In the following, each converter is briefly analyzed.

A. Charger: Buck-boost converter

Since the battery charger is adopted with buck-boost converter, its key parameters include duty ratio $D_{11}$ and inductor $L_m$. Therefore, duty ratio $D_{11}$ and inductor $L_m$ are derived in the following.

A.1 Duty ratio $D_{11}$

In the light day, the proposed hybrid converter is regarded as the charger. The power flows from PV arrays to battery. During switching cycle, battery voltage $V_B$ is almost kept at a constant value. For maximum power point tracking (MPPT) of solar power, the proposed one can regulate the charging current $I_B$ to implement MPPT. The maximum duty ratio $D_{11(max)}$ can be determined under the minimum output voltage $V_{PV(min)}$ of solar power and maximum battery voltage $V_{B(max)}$. Its relationship is expressed as

$$V_{PV(min)}D_{11(max)}T_s + \left(-V_{B(max)}\right)\left(1 - D_{11(max)}\right)T_s = 0, \tag{1}$$

where $T_s$ represents the period of the proposed hybrid converter. From (1), $D_{11(max)}$ can be derived by

$$D_{11(max)} = \frac{V_{B(max)}}{V_{PV(min)} + V_{B(max)}}. \tag{2}$$

In addition, maximum transfer ratio $M_{11(max)}$ can be obtained as

$$M_{11(max)} = \frac{D_{11(max)}}{1 - D_{11(max)}}. \tag{3}$$

According to the above equations, when type of battery is selected, the maximum charging current $I_{B(max)}$ can be denoted. Moreover, the charging current $I_B$ can changed

from its maximum charging current $I_{B(max)}$ to 0A by regulating duty ratio $D_{11}$ of switch $M_1$. The charging current $I_B$ is determined by MPPT of solar power.

A.2 Inductor $L_m$

In order to obtain the inductance $L_m$, the boundary inductance $L_{mB}$, which is the inductor value of the proposed converter operated in the boundary of CCM and discontinuous condition mode (DCM). Its conceptual waveforms are illustrated in Figure 12. The average charging current $I_{B(av)}$ can be determined as

$$I_{B(av)} = \frac{\Delta I_{Lm(max)}(1 - D_{11})}{2}, \tag{4}$$

where $\Delta I_{Lm(max)}$ represents a maximum current variation of inductor $L_m$. In (4), $\Delta I_{Lm(max)}$ can be expressed by

$$\Delta I_{Lm(max)} = \frac{V_{PV}D_{11}T_s}{L_{mB}} = \frac{V_B(1 - D_{11})T_s}{L_{mB}} \tag{5}$$

where $V_{PV}$ is the output voltage of solar power and $V_B$ represents the battery voltage. According to (4) and (5), the charging current $I_{B(av)}$ can be obtained as

$$I_{B(av)} = \frac{(1 - D_{11})^2 V_B T_s}{2L_{mB}}. \tag{6}$$

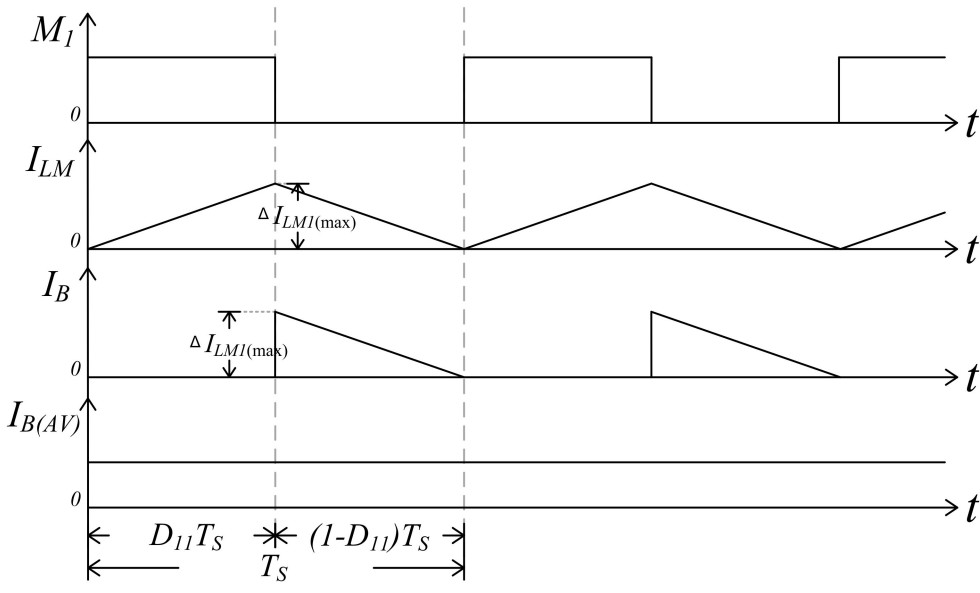

**Figure 12.** Conceptual waveforms of inductor current $I_{Lm}$ and charging current $I_B$ in the charger.

Since the maximum charging current $I_{B(av)max}$ occurs at the maximum battery voltage $V_{B(max)}$ and the minimum PV voltage $V_{PV(min)}$, the maximum charging current $I_{B(av)max}$ can be rewritten with

$$I_{B(av)max} = \frac{(1 - D_{11(min)})^2 V_{B(max)} T_s}{2L_{mB}}, \tag{7}$$

where $D_{11(min)}$ represents duty ratio from $V_{B(max)}$ to $V_{PV(max)}$. Since the charger is always operated in CCM, $I_{B(av)max}$ can be expressed by $K_1 I_{B(max)}$, where $K_1$ varies from 0 to 1 and

$I_{B(max)}$ is the maximum charging current. In general, $K_1$ is set at 0.1~0.3. From (7), it can be seen that inductor $L_{m1}$ can be expressed as

$$L_{m1} = \frac{(1 - D_{11(min)})^2 V_{B(max)} T_s}{2K_1 I_{B(max)}}.$$　　(8)

A.3. selection of switches

Figure 7a shows the schematic diagram of the proposed hybrid converter operated in the charging mode. In order to determine voltage and $V_{PV}$ is at the maximum value and battery voltage $V_B$ is at maximum value, voltage ratings of components in the proposed one can be determined. Maximum voltage stresses of $M_1$ and $M_2$ can be determined by

$$V_{M1(max)} = V_{M2(max)} = V_{PV(max)} + V_{B(max)}.$$　　(9)

In addition, voltage stress of switch $S_1$ obtained as

$$V_{S1(max)} = N V_{B(max)}$$　　(10)

When input voltage $V_{pV}$ is at minimum value and battery voltage $V_B$ is at maximum value, the maximum *rms* current $I_{M(rms)}$ of switch $M_1$ can be illustrated by

$$I_{M1(rms)} = \frac{I_{B(max)}}{1 - D_{11(max)}} \sqrt{D_{11(max)} [1 + \frac{r^2}{12}]},$$　　(11)

where $r$ is defined by $(\frac{\Delta I_{Lm}}{I_{B(max)}})$. The maximum *rms* current $I_{M2(rms)}$ of switch $M_2$ can be derived as

$$I_{M2(rms)} = I_{B(max)} \sqrt{\frac{[1 + \frac{r^2}{12}]}{1 - D_{11(max)}}}.$$　　(12)

Moreover, the maximum *rms* current $I_{LK(rms)}$ of inductor $L_M$ can be obtained as

$$I_{LK(rms)} = \frac{I_{B(max)}}{1 - D_{11(max)}} \sqrt{1 + \frac{r^2}{12}}.$$　　(13)

Discharger: Active clamp flyback converter

When the proposed boost converter is operated in the discharging mode, its equivalent circuit is composed by flyback converter with the active clamp circuit. For design of active clamp flyback converter, the important parameters include duty ratio $D_{12}$, transformer $T_r$, active clamp capacitor $C_C$ and output capacitor $C_o$. In the following, their designs are analyzed briefly.

B.1. Duty ratio $D_{12}$

When the proposed hybrid converter uses flyback converter with the active clamp circuit to achieve soft-switching features, the active clamp circuit does not affect transfer ratio $M_{12}$ of the proposed flyback converter. That is, transfer ratio $M_{12}$ is the same as the conventional one. According to volt-second balance of inductor $L_m$, the following equation can be obtained by

$$V_B D_{12} T_s + \left(-\frac{V_o}{N}\right)(1 - D_{12}) T_s = 0,$$　　(14)

where $N$ $(=N_2/N_1)$ is the turns ratio of transformer $T_r$. From (9), it can be found that transfer ratio $M_{12}$ can be represented as

$$M_{12} = \frac{N D_{12}}{(1 - D_{12})}.$$　　(15)

When the output to input voltage transfer ratio $M_{12}$ is determined, duty ratio $D_{12}$ can obtained by

$$D_{12} = \frac{V_o}{NV_B + V_o}. \tag{16}$$

In the (11), when $N$, $V_o$ and $V_B$ are specified, duty ratio $D_{12}$ can be determined.

B.2. Transformer $T_r$

In order to Design transformer $T_r$, turns ratio $N$ and the magnetizing inductor $L_m$ are important parameters. Since output current $I_o$ can be determined by inductance $L_m$ and turns ratio $N$, their conceptual waveforms is shown in Figure 13. From Figure 13, it can be found that the average diode current $I_{D1(av)}$ is represented by

$$I_{D1(av)} = \frac{\Delta I_{Lm2(max)}(1 - D_{12})}{2N}, \tag{17}$$

where $\Delta I_{Lm2(max)}$ is the variation value of inductor current $I_{Lm2}$. According to operational principle of the proposed hybrid converter, inductor current $\Delta I_{Lm2(max)}$ is obtained with

$$\Delta I_{Lm2(max)} = \frac{V_{B(max)}D_{12}T_s}{L_{MB2}}, \tag{18}$$

where $L_{MB2}$ is the magnetizing inductance of transformer $T_r$ where the proposed hybrid converter is operated in the boundary of DCM and CCM.

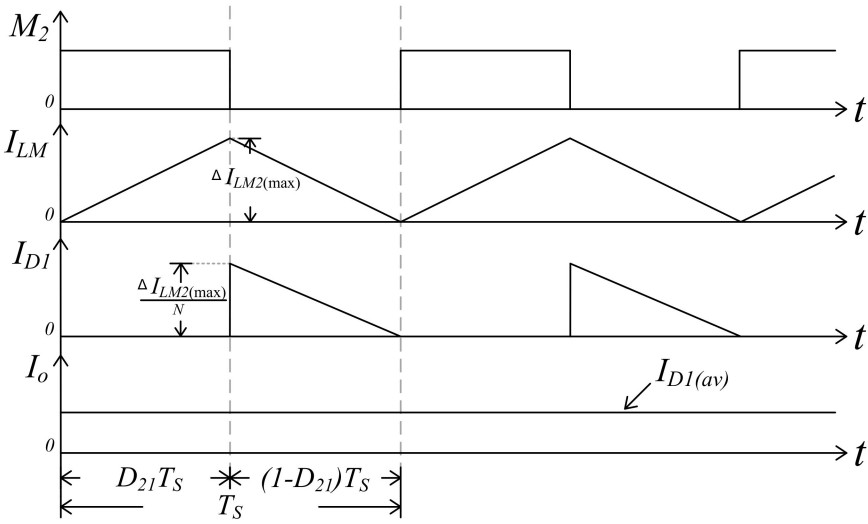

**Figure 13.** Conceptual waveforms of inductor current $I_{Lm}$ and output current $I_o$ in the discharger.

Since the proposed one adopts the active clamp circuit to achieve soft-switching features, its magnetizing inductor $L_{m2}$ is always operated in CCM. Therefore, the proposed one is designed in CCM under light load condition. The average current $I_{D1(av)}$ is equal to $K_2 I_{o(max)}$, where $K_2$ range from 0 to 1 and $I_{o(max)}$ represents the maximum output current. According to (12) and (13), the magnetizing inductor $L_{m2}$ can be determined by

$$L_{m2} = \frac{V_{B(max)}D_{12}(1 - D_{12})T_s}{2NK_2 I_{o(max)}}. \tag{19}$$

Since the magnetizing inductor $L_m$ is separately operated in the charging and discharging modes, their inductances are derived with different values ($L_{m1}$ and $L_{m2}$), respectively. In order to design a proper inductance $L_m$, it is selected with the maximum value between $L_{m1}$ and $L_{m2}$.

In (10) and (11), when voltage $V_B$ and $V_o$ are specified, turns ratio $N$ is inversely proportional to duty ratio $D_{12}$. Since a large duty ratio $D_{12}$ corresponds to a smaller turns ratio $N$ of transformer $T_r$. That is, lower current stresses are imposed on switches $M_1$ and $M_2$. However, in order to tolerate variations of load, battery voltage and component value, it is better to selected an operating ranges as $D = 0.35\sim0.4$. When duty ratio $D_{12}$ is specified, turns ratio $N$ can be determined.

B.3. Active clamp capacitor $Cc$

When the proposed hybrid converter adopts the active clamp to achieve soft-switching features, the active clamp capacitor $C_C$ can be used to recover energy trapped in leakage inductor $L_k$ and help switch to achieve *ZVS* features. In order to obtain a wider range of soft-switching features, a half of resonant period is equal to or greater than turn-off time of switch $M_2$ when capacitor $Cc$ and leakage inductor $L_k$ are formed as the resonant network. Therefore, capacitor $Cc$ must satisfy the following inequality:

$$\pi\sqrt{L_k C_c} \geq (1 - D_{12})T_s. \tag{20}$$

According to (15), capacitor $C_C$ can be expressed by

$$C_o \geq \frac{(1 - D_{12})^2 T_s{}^2}{\pi^2 L_k}. \tag{21}$$

In (10), once leakage inductor $L_k$ is specified, capacitor $C_C$ can be determined.

B.4. Output Capacitor $C_o$

Since output capacitor $C_o$ is used to reduce ripple of output voltage $V_o$, its value must be large enough. The ripple voltage $\Delta V_o$ across output capacitor $C_o$ is expressed as follows:

$$\Delta V_o = \frac{I_{o(max)} D_{12} T_s}{C_o}, \tag{22}$$

where $I_{o(max)}$ is the maximum output current. Therefore, output capacitor $C_O$ can be determined by

$$C_o = \frac{I_{o(max)} D_{12} T_s}{\Delta V o}. \tag{23}$$

When the maximum output current $I_{o(max)}$, duty ratio $D_{12}$, switching cycle $T_S$ and output ripple voltage $\Delta V_o$ are specified, output capacitor $C_o$ can be determined by (18).

B.4. Selection of switches and diode

Figure 7a shows the schematic diagram of the proposed hybrid converter operated in the discharging mode. When battery voltage $V_B$ is under a maximum value situation, voltage rating of components can be determined. Maximum voltage stresses of switch $M_1$ and $M_2$ can be obtained as

$$V_{M1(max)} = V_{M2(max)} = V_{B(max)} + \frac{V_o}{N}. \tag{24}$$

Maximum voltage stress $V_{D1(max)}$ of diode $D_1$ can be expressed by

$$V_{D1(max)} = N\,V_{B(max)} + V_o. \tag{25}$$

When the minimum battery voltage $V_{B(min)}$ and output maximum current $I_{o(max)}$, the maximum *rms* current $I_{M2(rms)}$ can be derived as

$$I_{M2(rms)} = \frac{N\,I_{o(max)}}{1 - D_{12(max)}}\sqrt{D_{12(max)}\left[1 + \frac{r^2}{12}\right]}. \tag{26}$$

The maximum *rms* current $I_{LM2(rms)}$ is expressed by

$$I_{M2(rms)} = \frac{N \, I_{o(max)}}{1 - D_{12(max)}} \sqrt{1 + \frac{r^2}{12}}.$$ (27)

Moreover, the maximum *rms* current $I_{S1(rms)}(= I_{D1(rms)})$ is indicated by

$$I_{S1(rms)} = I_{D1(rms)} = I_{o(max)} \sqrt{\frac{[1 + \frac{r^2}{12}]}{(1 - D_{12(max)})}}.$$ (28)

Since switch $M_2$ is turned off, inductor $L_K$ and capacitor $C_c$ form a resonant network. A half resonant period of the resonant network is equal to $(1 - D_{12}) \, T_s$. The current waveform of switch $M_1$ varies with cosine wave manner. According to the *rms* calculation method for the cosine wave, the maximum *rms* current $I_{M1(rms)}$ can be obtained by

$$I_{M1(rms)} = I_{PK} \sqrt{\frac{1 - D_{12(max)}}{2}},$$ (29)

where $D_{12(max)}$ is cut the minimum battery voltage $V_{B(min)}$ and $I_{PK}$ expresses the maximum current of inductor $L_{M2}$. When the proposed hybrid converter is operated in the heavy load condition, current $I_{PK}$ is approximately equal to $[N \, I_{o(max)} \, / (1 - D_{12(max)}) + \frac{1}{2} \Delta I_{Lm2}]$.

B.5. Power losses analysis

Since the proposed hybrid converter is operated in the charging mode, the proposed one is operated with hard-switching manner. Its power loss analysis is the same as the conventional buck converter.

The power loss analysis is neglected in this paper. When the proposed one is operated in the discharging mode, the active clamp capacitor $C_c$ can be used to recover the energy strapped in leakage inductor $L_k$ to increase conversion efficiency of the proposed one. Therefore, the power loss analysis is described for the proposed one operated in the discharging mode. When the proposed one is operated in the discharging mode, power loss includes losses of switches, diode and core. In the following, power loss analysis is derived.

(1). Losses of switches

The losses of switches include switching loss and conduction loss. Figure 14 shows the conceptual waveforms of switching losses for switches $M_1$ and $M_2$. Since switches $M_1$ and $M_2$ is operated with *ZVS* at turn-on transition, their switching loss is only induced at turn-off transition of switches. Therefore, switching losses $P_{soff}$ of switches $M_1$ and $M_2$ can be expressed by

$$P_{soff} = \frac{1}{2T_s} V_{M1(max)} (t_{off} \, I_{DP}),$$ (30)

where $I_{DP}$ is equal to $[N \, I_{o(max)} \, / (1 - D_{12(max)}) + \frac{1}{2} \Delta I_{Lm2}]$. The conduction loss of switch $M_1$ ( or $M_2$) can be derived as

$$P_{CD} = I_{M(rms)}^2 \, R_{DS(on)},$$ (31)

where $I_{M(rms)}$ is the *rms* current of each switch and $R_{DS(on)}$ represses a resistance of switch during turn-on state.

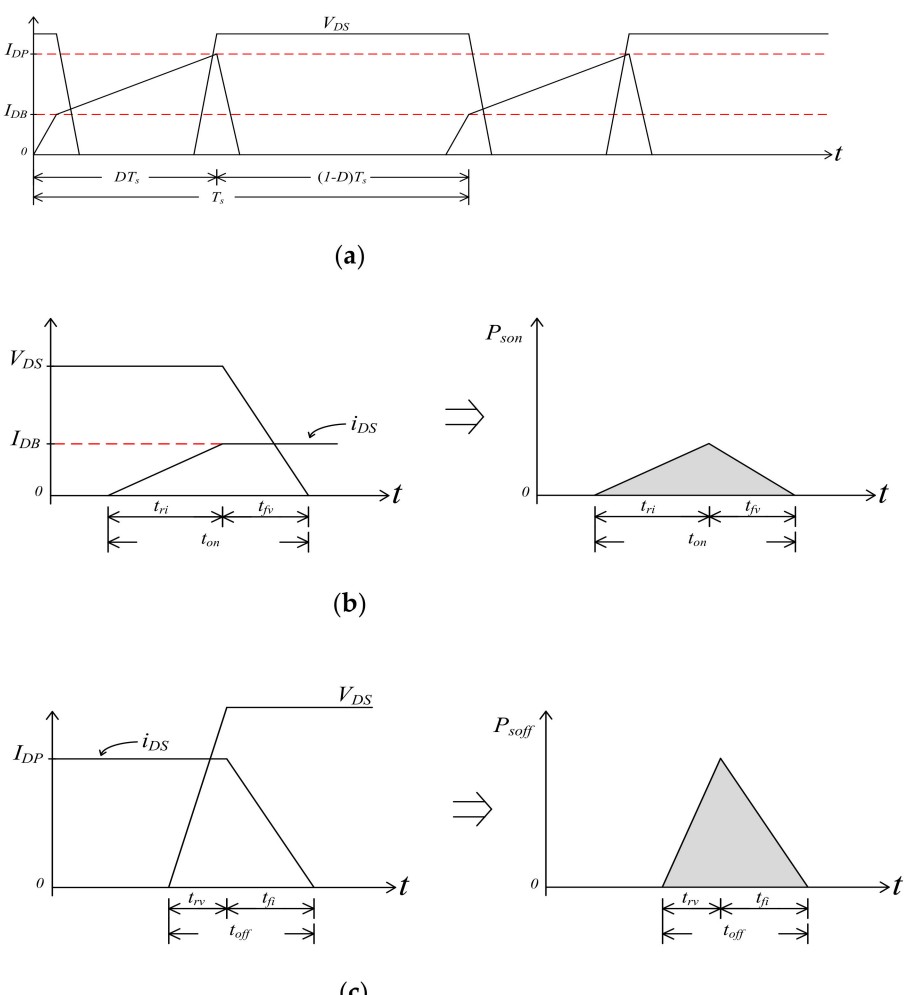

**Figure 14.** Conceptual waveforms of switching loss during switch turn-on and turn-off transitions (**a**) during one switching cycle, (**b**) during turn-on transition (**c**) during turn-off transition.

(2). Loss of diode

The loss of diode $D_1$ is generated by the forward voltage $V_F$ when diode $D_1$ is in the forward biased state. The loss $P_{D1}$ can be derived by

$$P_{D1} = I_{D1(rms)}{}^2 V_F. \tag{32}$$

(3). Loss of core

The loss of core includes core loss and copper loss. The core loss of transformer $T_r$ is determined by the maximum flux density $B_m$ and core loss curve of core. The maximum flux density $B_m$ can be determined by

$$B_m = \frac{\mu_o \mu_r N \left( \frac{N\, I_{o(max)}}{1 - D_{12(max)}} + \frac{\Delta I_{Lm2}}{2} \right)}{(l_e + \mu_r l_g)}, \tag{33}$$

where $N$ is the turns of primary winding and $l_e$ expresses the effective magnetic path length, $l_g$ indicates air gap length and $u_r$ permeability. When $B_m$ is determined, the core loss coefficient $C_p$ can be obtained through core loss curve of core. The core loss $P_{CL}$ is determined as

$$P_{CL} = C_P V_e, \tag{34}$$

where $V_e$ is the effective core volume of core. Moreover, copper loss $P_{CPL}$ can be derived by

$$P_{CPL} = I_{Lm2(rms)}{}^{2}\, R_{dc1}\, l_{m1} + I_{D1(rms)}{}^{2}\, R_{dc2}\, l_{m2},\tag{35}$$

where $R_{dc1}$ is the resistance coefficient of wire gauge of primary winding, $l_{m1}$ represses the total length of turns of primary winding, $R_{dc2}$ is the resistance coefficient of wire gauge of secondary winding and $l_{m2}$ indicates the total length of turns of secondary winding.

A. Block diagram of control method of the proposed hybrid converter

In order to control the proposed hybrid converter, a microcontroller and pulse-width modulation integrated circuit (PWM IC) are adopted in the proposed systems, as shown in Figure 15. In Figure 15, the microcontroller is used to implement maximum power point tracking (MPPT) of solar power, manages battery charging, controls battery charging current and perform battery protection. Moreover, the PWM IC is adopted to regulate output voltage $V_o$. For MPPT, this paper uses the perturb-and-observe method to execute the MPPT of solar power. In order to match the MPPT of solar power, charger adopts constant current (CC) method to implement battery charging.

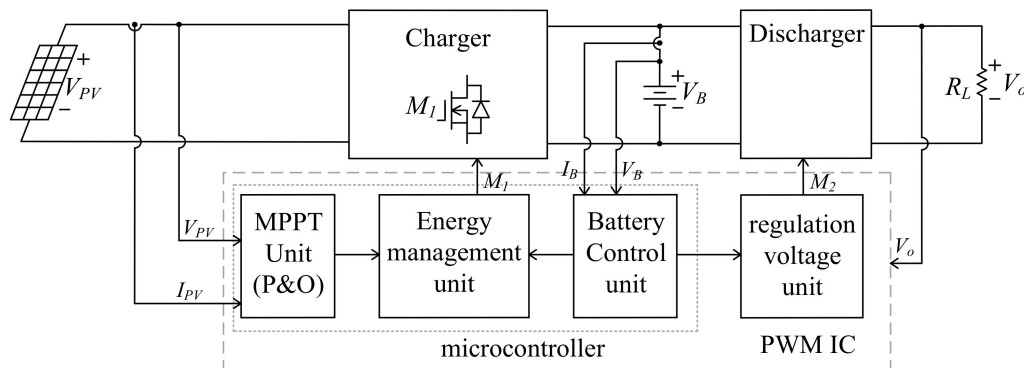

**Figure 15.** Block diagram of the proposed hybrid converter for solar power system applications.

B. Performances comparison between the proposed hybrid converter and the conventional counterpart converter

In general, key components of switching power supply include the switch, diode, magnetic device, capacitor, printed circuit board (PCB), control IC, driving circuit, filter and resistor, and so on. According to the technical report of the Industrial Economics and Knowledge Center (IEK) in Taiwan, cost of each component in switching power supply is illustrated in Table 1. From Table 1, it can be found that switch, capacitor, magnetic device, diode and driving circuit possess higher cost ratio in the switching power supply. Table 2 lists the component counts comparison between the proposed hybrid converter and the conventional counterpart converter. Since the conventional counterpart converter shown in Figure 4 includes three switches, two magnetic devices, two diodes and two sets of driving circuits, the proposed hybrid converter can reduce component usage and increase an extra switch $S_1$ usage. When the proposed one reduces one switch usage, it can obtain a cost reduction of 6.7%. In addition, the reduction magnetic device, diode and driving circuit usage of the proposed one can acquire a cost reduction of 8%, 5% and 6%, respectively. In order to reduce component counts, the proposed one increases a cost of 3~6.7%. From Table 2, it can be found that the proposed one can reduce cost of 19–22.7%.

**Table 1.** Total cost analysis of switching power supply (data from Industrial Economics and Knowledge Center (IEK) in Taiwan).

| Component | Cost Ratio = (Cost ofEach Item/Total Cost) |
|---|---|
| switch | 20% |
| capacitor | 18% |
| Magnetic device | 16% |
| diode | 10% |
| PCB | 6% |
| filter | 8% |
| Driving circuit | 12% |
| resistor | 5% |
| Control IC | 3% |
| others | 2% |

**Table 2.** Component count comparison between the proposed hybrid converter and the conventional counterpart converter.

| Component | The Conventional Hybrid Buck-Boost/Flyback Hybrid Converter (Figure 4) | The Proposed Hybrid Buck-Boost/Flyback Hybrid Converter (Figure 5) | Cost Ratio decrEase for the Proposed Hybrid Converter |
|---|---|---|---|
| switch | 3 pcs | 2 pcs | 6.7% |
| Magnetic device | 2 pcs | 1 pc | 8% |
| diode | 2 pcs | 1 pc | 5% |
| capacitor | 3 pcs | 3 pcs | 0% |
| Driving circuit | 2 sets | 1 sets | 6% |
| Extra device | 0 pc | 1 pc | $-3 \sim -6.7\%$ |

## 5. Experimental Results

The proposed hybrid converter used solar power as its input source. Specifications of solar power are listed in Table 3. The following specifications were implemented.

**Table 3.** Specification of the solar power supplied by solar power manufacturer.

| Parameter | Values |
|---|---|
| Maximum power ($P_{max}$) | 30 W |
| Maximum power voltage ($P_{pv(max)}$) | 17.5 V |
| Maximum power current ($I_{pv(max)}$) | 1.84 A |
| Open circuit voltage ($V_{oc}$) | 20.9 V |
| Short circuit current ($I_{sc}$) | 1.94 A |

A. Charger: Buck-boost converter

- Input voltage $V_{PV}$: DC 17.5 V~20.6 V (solar power),
- Switching frequency $f_{s1}$: 50 kHz,
- Output voltage $V_B$: DC 8 V~12 V (lithium battery:3.2 Ah), and
- Maximum charging current $I_{B(max)}$: 3.2 Ah

B. Discharger: flyback converter

- Input voltage $V_B$: DC 8 V~12 V (lithium battery:3.2 Ah),
- Switching frequency $f_{s2}$: 50 kHz,
- Output voltage $V_o$: DC 10 V, and

- Maximum output current $I_{o(max)}$: 2 A.

According to the previously specifications and design of the proposed hybrid converter, inductor $L_m$, turns ratio $N$ and active clamp capacitor $C_C$ could be determined. Table 4 illustrates parameters of components of the proposed hybrid converter. According to operational conditions of the proposed one, current and voltage stresses could be determined. From Table 4, it can be obtained that the magnetizing inductor $L_m$ equaled 660 μH and turns ratio $N$ was equal to 2. When transformer $T_r$ was wound with the magnetizing inductor $L_m$ of 660 μH, leakage inductor $L_k$ was measured and its value was 12.5 μH. Therefore, the active capacitor $C_C$ was calculated and its value was 1.62 μF. Capacitor $C_C$ is adopted with 1.5 μF. In addition, current and voltage stresses of the proposed hybrid converter and the conventional counterpart converter are listed in Table 5. From Table 5, it can be found that although component stress of the proposed hybrid converter was higher than that of the conventional counterpart converter, it could use fewer component counters to achieve the charging and discharging functions. The switch $S_1$ could adopt lower current and voltage stresses to change the charging mode or discharging mode of the proposed one. Furthermore, the components of power stage in the proposed hybrid converter were determined as follows:

- Switches $M_1$, $M_2$: AoW2918,
- Diode $D_1$: STPS10L60D,
- Switches $S_1$: AoW2918,
- Transformer $T_r$: EE-33 core, and
- Output capacitor $C_o$: 47 μF/25 V.

Figure 16 shows the photo of the proposed hybrid prototype converter. The hardware dimension of the proposed hybrid converter was about $100 \times 60$ mm$^2$. The circuit layout safety distance was set at 5 mm around outside of each component. According to the requirement of safety distance of each component, circuit layout area comparison between the proposed hybrid converter and the conventional counterpart converter is listed in Table 6. When switch $M_1$ was adopted with AoW2918, its package wasTO220. According to dimension of TO220 package, component dimension was $10 \times 5$ mm$^2$. In order to consider safety distance between two components, circuit layout dimension was considered with $20 \times 15$ mm$^2$. Although a component with a heat sink could increase its power processing capacity, its circuit layout dimension was increased. The heat sink dimension for TO220 package was $15 \times 10$ mm$^2$. Therefore, switch with heat sink needed $25 \times 20$ mm$^2$ for switch layout dimension. According to the above requirement to layout the proposed hybrid converter and the conventional counterpart converter, their circuit layout area is respectively calculated in the Table 6. From Table 6, it can be found that the proposed hybrid converter needed a circuit layout area of 6000 mm$^2$, while the conventional counterpart converter needed that of 9000 mm$^2$. Therefore, the proposed one could reduce circuit layout area by 3000 mm$^2$. The power density of the proposed one could increase about 1.5 times.

The proposed hybrid converter used solar power to charge the battery with CC. The MPPT and battery charging with CC must be implemented. When solar power was used as input voltage source, the proposed hybrid converter adopted the perturb-and-observe method to implement MPPT. Measured voltage $V_{PV}$, current $I_{PV}$, and power $P_{PV}$ waveforms of solar power is shown in Figure 17. Figure 17a illustrates those waveforms under $P_{PV(max)} = 15$ W, while Figure 17b plots those waveforms under $P_{PV(max)} = 30$ W. From Figure 17, it can be obtained that tracking time $T$ of solar power was about 200 ms. Figure 18 shows measured gate voltage $M_1$ of switch $M_1$ and charging current $I_B$. Since capacitor $C_C$ connected with inductor $L_m$ in series, they formed a resonant network through battery or capacitor $C_{DC}$ during switch $M_1$ turn-on or turn-off interval, respectively. The measured charging current $I_B$ varied with a resonant waveform. Figure 18a shows those waveforms under the average charging current $I_{B(av)} = 0.7$ A. Moreover, Figure 18b illustrates those waveforms under $I_{B(av)} = 3$ A.

**Table 4.** Parameters of components in the proposed hybrid converter.

| Operational Mode | Design Value | | | | Practical Value (v) | Conditions |
|---|---|---|---|---|---|---|
| | Symbol | Parameter | Equation | Value | | |
| The charging mode | Switch $M_1$ | $V_{M1(max)}$ | (9) | 33 V | 33 V | $V_{B(max)}=$ 12 V $V_{PV(max)}=$ 20.6 V $D_{11(max)}=$ 0.6 $D_{11(min)}=$ 0.368 $\Delta I_{LM}=$ 0.64 A, $r=0.2$ |
| | | $I_{M1(rms)}$ | (11) | 6.22 A | 6.20 A | |
| | | $I_{M1(pk)}$ | | 8.32 A | 8.07 A | |
| | Switch $M_2$ | $V_{M2(max)}$ | (9) | 33 V | 33 V | $V_{B(max)}=$ 12 V $V_{PV(max)}=$ 20.6 V $V_{PV(min)}=$ 8 V $D_{11(max)}=$ 0.6 $D_{11(min)}=$ 0.368 $\Delta I_{Lm}=$ 0.64 A, $r=0.2$ |
| | | $I_{M2(rms)}$ | (12) | 5.07 A | 5.06 A | |
| | | $I_{M2(pk)}$ | | 8.32 A | 8.07 A | |
| | Switch $S_1$ | $V_{S1(max)}$ | (10) | 24 V | 24 V | N = 2 $V_{B(max)}=$ 12 V |
| | | $I_{S1(rms)}$ | | 0 A | 0 A | |
| | | $I_{S1(pk)}$ | | 0 A | 0 A | |
| | Inductor $L_{m1}$ | $L_{m1}$ | (8) | 150 μH | 660 μH $\Delta I_{Lmp}=$ 0.145 A $r_p=$ 0.045 $I_{Lk(rms)}=$ 8 A | $D_{11(min)}=$ 0.368, $D_{11(max)}=$ 0.6 $K=0.1$, $T_s=$ 20 μS, $V_{B(max)}=$ 12 V, $I_{B(max)}=$ 3.2 A $\Delta I_{LM}=$ 0.64 A, $r=0.2$ |
| | | $I_{LK(rms)}$ | (13) | 8.01 A | | |
| The discharging mode | Switch $M_1$ | $V_{M1(max)}$ | (24) | 17 V | 17 V | $V_{B(max)}=$ 12 V, $V_{B(min)}=$ 8 V $D_{12(min)}=$ 0.294, $D_{12(max)}=$ 0.385 N = 2, $V_o=$ 10 V, $r=0.2845$ $I_{o(max)}=$ 2 A |
| | | $I_{M1(rms)}$ | (29) | 3.92 A | 3.63 A | |
| | | $I_{M1(pk)}$ | | 7.07 A | 6.56 A | |
| | Switch $M_2$ | $V_{M2(max)}$ | (24) | 17 V | 17 V | $V_{B(max)}=$ 12 V $D_{12(max)}=$ 0.385 N = 2, $V_o=$ 10 V, $r=0.2845$ $I_{o(max)}=$ 2 A |
| | | $I_{M2(rms)}$ | (26) | 1.05 A | 4.04 A | |
| | | $I_{M2(pk)}$ | | 7.07 A | 6.56 A | |
| | Switch $S_1$ | $V_{S1(max)}$ | | 0 V | 0 V | $D_{12(max)}=$ 0.385 $r=0.2845$ $I_{o(max)}=$ 2 A |
| | | $I_{S1(rms)}$ | (28) | 2.56 A | 2.55 A | |
| | | $I_{S1(pk)}$ | | 3.54 A | 3.28 A | |
| | Transformer $T_r$ | $L_{m2}$ | (19) | 62 μA | 660 μH $\Delta I_{Lmp}=$ 0.107 A $r_p=$ 0.027 $I_{Lm2(rms)}=$ 6.5 A | $V_{B(max)}=$ 12 V, $D_{12}=$ 0.294 $K_2=0.1$, $T_s=$ 20 μS, N = 2 $I_{o(max)}=$ 2 A $\Delta I_{Lm2}=$ 1.138 A, $r=0.2845$ |
| | | $I_{Lm(rms)}$ | (27) | 6.53 A | | |
| | Capacitor $C_c$ | $C_c$ | (21) | 1.62 μF | 1.5 μF | $L_k=$ 12.5 μH, $D_{12(min)}=$ 0.294 $T_s=$ 20 μS |

**Table 5.** Current and voltage stresses of components between the proposed hybrid converter and the counterpart converter.

| Component | The counTerpart Converter Shown in Figure 4 | | | The Proposed Hybrid Converter Shown in Figure 5 | | |
|---|---|---|---|---|---|---|
| | Symbol | Voltage Stress | Current Stress (rms) | Symbol | Voltage Stress | Current Stress (rms) |
| switches | $M_1$ | 33 V | 6.20 A | $M_1$ | 33 V | 6.20 A |
| | $M_2$ | 17 V | 4.04 A | $M_2$ | 33 V | 5.06 A |
| | $M_3$ | 17 V | 3.63 A | $S_1$ | 24 V | 2.55 A |
| Diodes | $D_1$ | 33 V | 5.06 A | $D_1$ | 34 V | 2.55 A |
| | $D_2$ | 34 V | 2.55 A | | | |

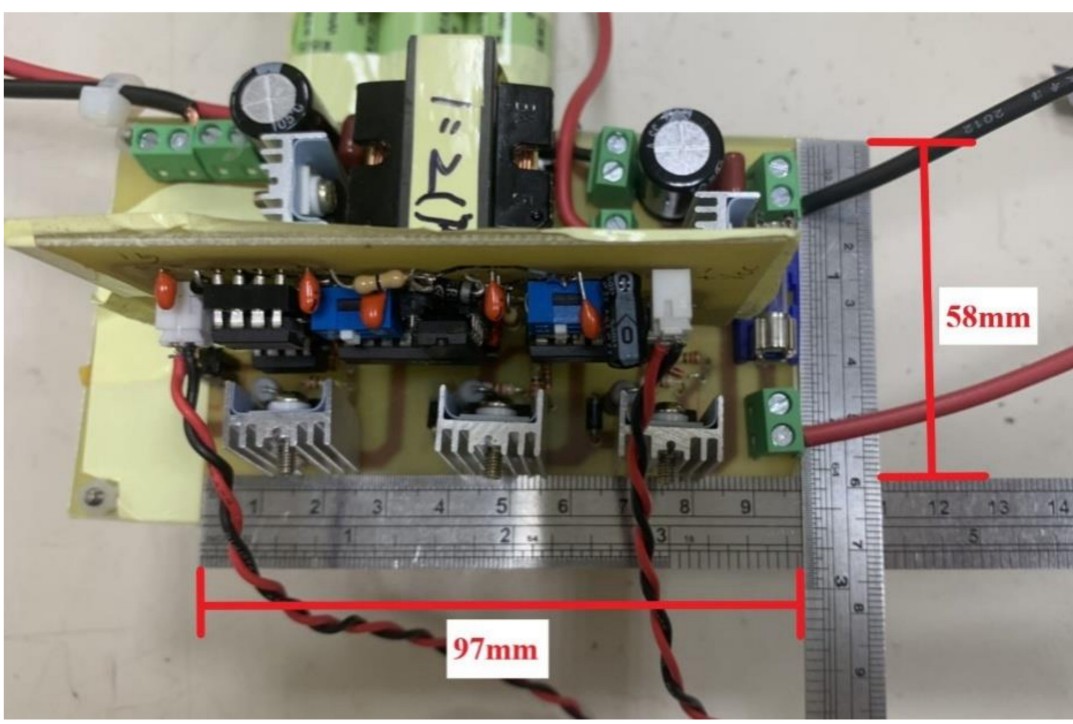

**Figure 16.** Photo of the proposed hybrid converter prototype.

**Table 6.** Layout area comparison between the propose hybrid converter and the conventional counterpart converter.

| Component | The Conventional Counterpart Converter Shown in Figure 4 | | | The Proposed Hybrid Converter Shown in Figure 5 | | |
|---|---|---|---|---|---|---|
| | Symbol | Dimension (Length × Width) | Circuit Layout Dimension (Length × Width) | Symbol | Dimension (Length × Width) | Circuit Layout Dimension (Length × Width) |
| Switches (head sink dimension Length × width = 15 mm × 10 mm) | Switch $M_1$ with heat sink | 10 mm × 5 mm $S_{M1} = 50$ mm$^2$ | 25 mm × 20 mm $S_{PM1} = 500$ mm$^2$ | Switch $M_1$ with heat sink | 10 mm × 5 mm $S_{M1} = 50$ mm$^2$ | 25 mm × 20 mm $S_{PM1} = 500$ mm$^2$ |
| | Switch $M_2$ with heat sink | 10 mm × 5 mm $S_{M2} = 50$ mm$^2$ | 25 mm × 20 mm $S_{PM2} = 500$ mm$^2$ | Switch $M_2$ with heat sink | 10 mm × 5 mm $S_{M2} = 50$ mm$^2$ | 25 mm × 20 mm $S_{PM2} = 500$ mm$^2$ |
| | Switch $M_3$ with heat sink | 10 mm × 5 mm $S_{M3} = 50$ mm$^2$ | 25 mm × 20 mm $S_{PM3} = 500$ mm$^2$ | Switch $S_1$ without heat sink | 10 mm × 5 mm $S_{s1} = 50$ mm$^2$ | 20 mm × 15 mm $S_{PS1} = 300$ mm$^2$ |
| Magnetics (EE-33 core) | Inductor $L_1$ | 35 mm × 35 mm $S_{L1} = 1225$ mm$^2$ | 45 mm × 45 mm $S_{PL1} = 2025$ mm$^2$ | Transformer $T_r$ | 35 mm × 35 mm $S_{Tr} = 1225$ mm$^2$ | 45 mm × 45 mm $S_{PTr} = 2025$ mm$^2$ |
| | Transformer $T_r$ | 35 mm × 35 mm $S_{Tr} = 1225$ mm$^2$ | 45 mm × 45 mm $S_{PTr} = 2025$ mm$^2$ | | | |
| Diodes | Diode $D_1$ with heat sink | 10 mm × 5 mm $S_{D1} = 50$ mm$^2$ | 25 mm × 20 mm $S_{PD1} = 500$ mm$^2$ | Diode $D_1$ with heat sink | 10 mm × 5 mm $S_{D1} = 50$ mm$^2$ | 25 mm × 20 mm $S_{PD1} = 500$ mm$^2$ |
| | Diode $D_2$ with heat sink | 10 mm × 5 mm $S_{D2} = 50$ mm$^2$ | 25 mm × 20 mm $S_{PD2} = 500$ mm$^2$ | | | |
| | Diode $D_B$ with heat sink | 10 mm × 5 mm $S_{DB} = 50$ mm$^2$ | 25 mm × 20 mm $S_{PDB} = 500$ mm$^2$ | Diode $D_B$ with heat sink | 10 mm × 5 mm $S_{DB} = 50$ mm$^2$ | 25 mm × 20 mm $S_{PDB} = 500$ mm$^2$ |
| others | Fuse, connector, Driving circuit and filter | $S_{PT} > 1675$ mm$^2$ | | Fuse, connector, Driving circuit and filter | $S_{PT} = 1675$ mm$^2$ | |
| Total circuit layout area $S_{TA}$ | | $S_{TA} > 8725$ mm$^2$ (=9000 mm$^2$) | | | $S_{TA} = 6000$ mm$^2$ | |

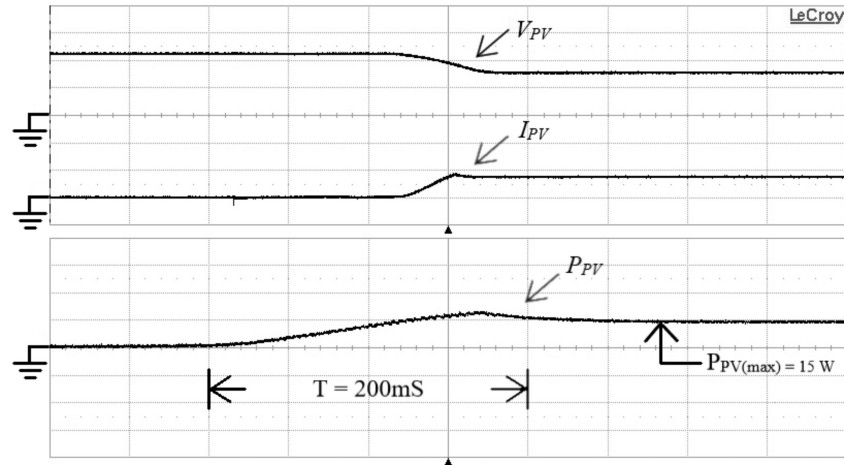

(*V$_{PV}$*: 10 V/div, *I$_{PV}$*: 1 A/div, *P$_{PV}$*: 15 W/div, time: 50 ms/div)

(**a**)

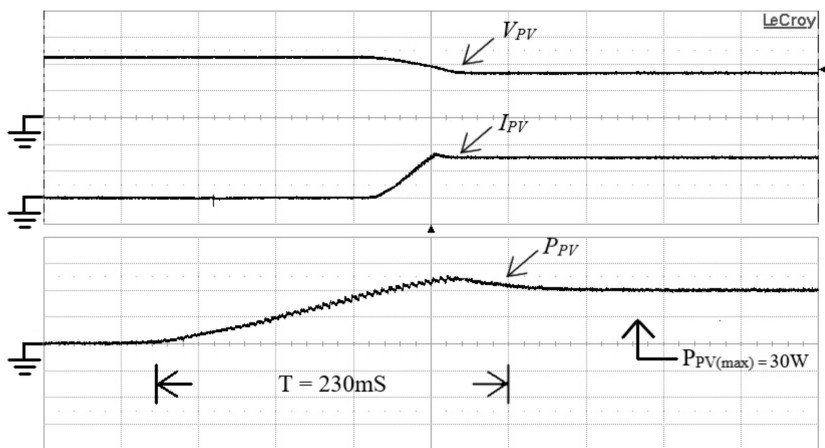

(*V$_{PV}$*: 10 V/div, *I$_{PV}$*: 1 A/div, *P$_{PV}$*: 15 W/div, time: 50 ms/div)

(**b**)

**Figure 17.** Measured voltage $V_{PV}$, current $I_{PV}$ and power $P_{PV}$ of solar power under maximum power point tracking (MPPT): (**a**) $P_{PV(max)}$ = 15 W, and (**b**) $P_{PV(max)}$ = 30 W.

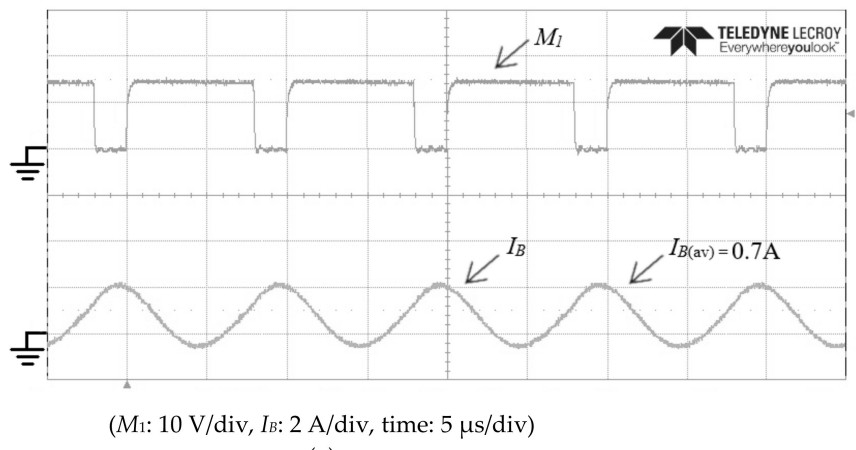

(*M$_1$*: 10 V/div, *I$_B$*: 2 A/div, time: 5 μs/div)

(**a**)

**Figure 18.** *Cont.*

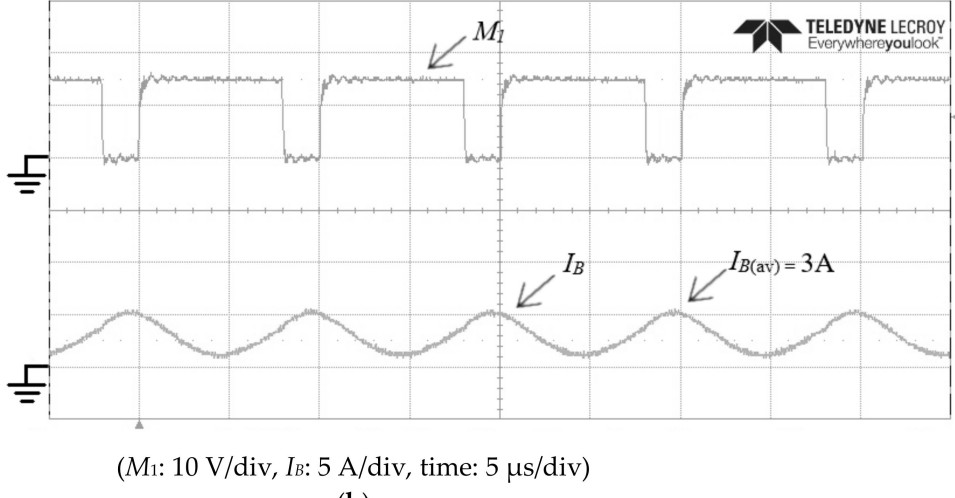

(*M*₁: 10 V/div, *I*ᵦ: 5 A/div, time: 5 μs/div)

(**b**)

**Figure 18.** Measured gate voltage $M_1$ and charging current $I_B$ of the proposed hybrid converter operated in the charging mode: (**a**) under the average charging current $I_{B(av)}$ = 0.7 A, and (**b**) under $I_{B(av)}$ = 3 A.

Since the proposed hybrid converter was operated in the discharging mode, switches $M_1$ and $M_2$ were operated with ZVS at turn-on transition. When the proposed hybrid converter was operated in the discharging mode, measured switch voltages $V_{M1}$, $V_{M2}$ and currents $I_{M1}$, $I_{M2}$ waveforms of the proposed hybrid converter are shown in Figure 19. Figure 19a,b show those waveforms under 10% of full-load condition, while Figure 20a,b illustrate those waveforms under 15% of full-load condition. From Figures 19 and 20, it can be found that switch $M_1$ and $M_2$ were operated with ZVS at turn-on transition under 10–15% of full-load condition, simultaneously. Figure 21 illustrates measured output voltage $V_o$ and output current $I_o$ under step-load change between 0% of full-load and 100% of full-load conditions, from which it can be obtained that the voltage regulation of output voltage $V_o$ was limited within ±1%.

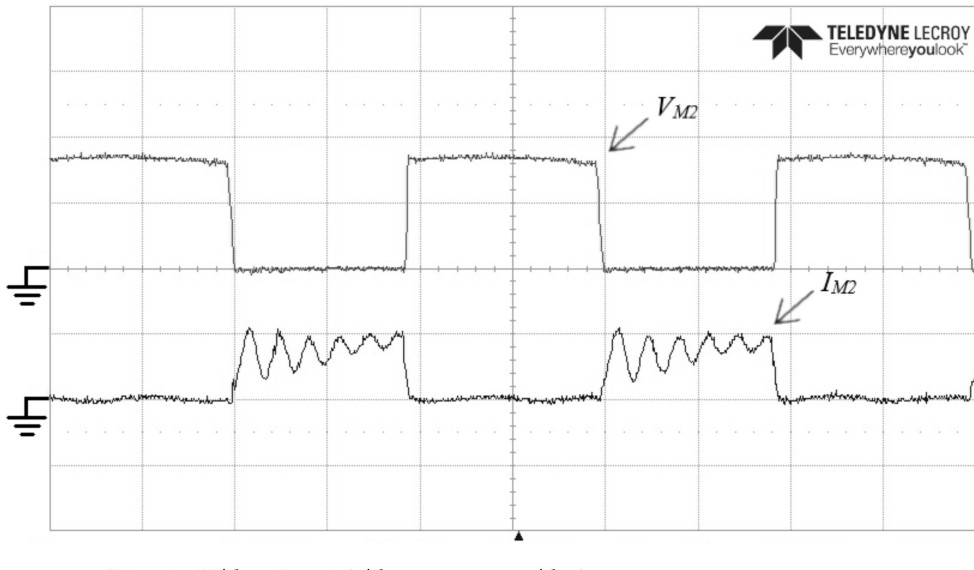

(*V*_{M2}: 10 V/div, *I*_{M2}: 1 A/div, time: 5 μs/div)

(**a**)

**Figure 19.** *Cont.*

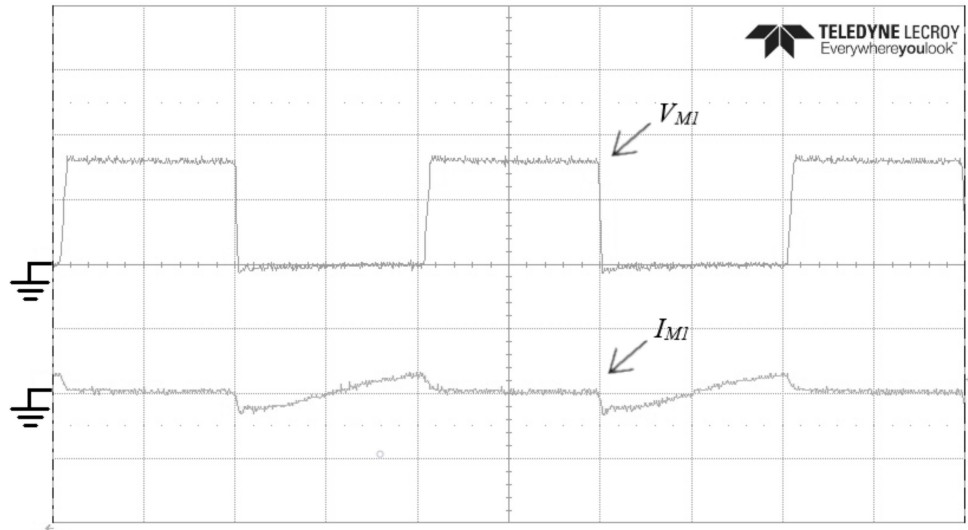

($V_{M1}$: 10 V/div, $I_{M1}$: 1 A/div, time: 5 µs/div)

(**b**)

**Figure 19.** Measured voltage $V_{M1}$, $V_{M2}$ and currents $I_{M1}$, $I_{M2}$ waveforms of the proposed hybrid converter operated in the discharging mode: (**a**) voltage $V_{M2}$ and current $I_{M2}$, and (**b**) voltage $V_{M1}$ and current $I_{M1}$ under 10% of full-load condition.

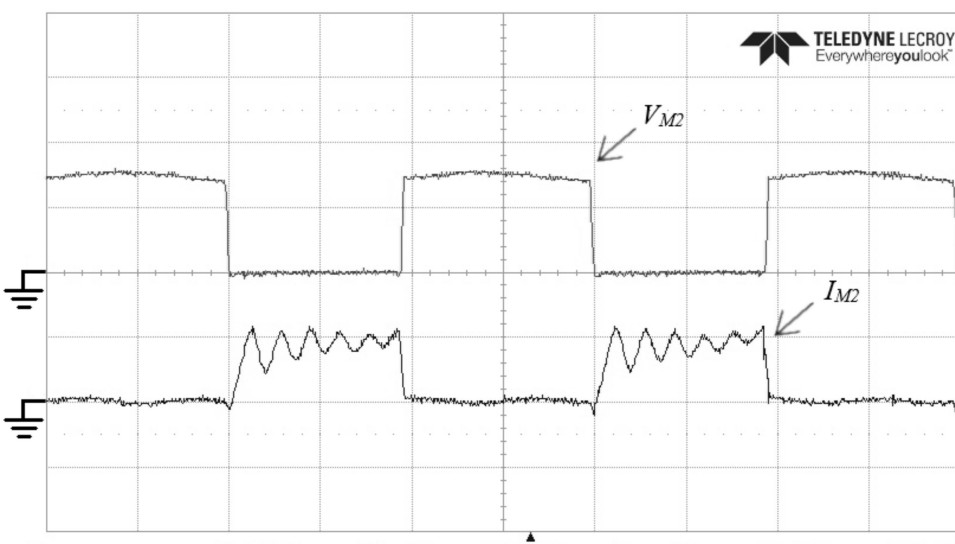

($V_{M2}$: 10 V/div, $I_{M2}$: 1 A/div, 5 µs/div)

(**a**)

**Figure 20.** *Cont.*

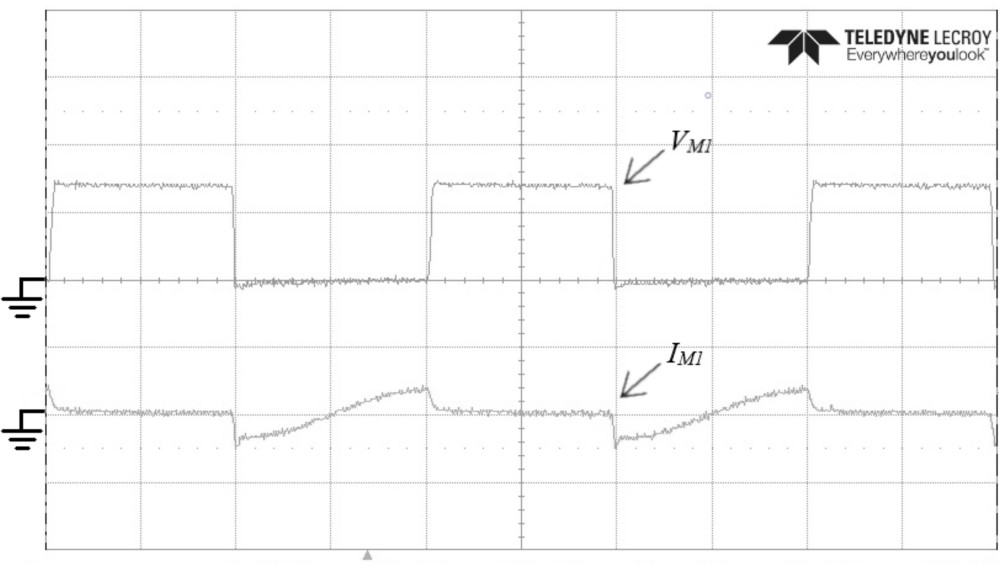

($V_{M1}$: 10 V/div, $I_{M1}$: 1 A/div, time: 5 μs/div)

(**b**)

**Figure 20.** Measured voltage $V_{M1}$, $V_{M2}$ and currents $I_{M1}$, $I_{M2}$ of the proposed hybrid converter operated in the discharging mode: (**a**) voltage $V_{M2}$ and current $I_{M2}$, and (**b**) voltage $V_{M1}$ and current $I_{M1}$ under 15% of full-load condition.

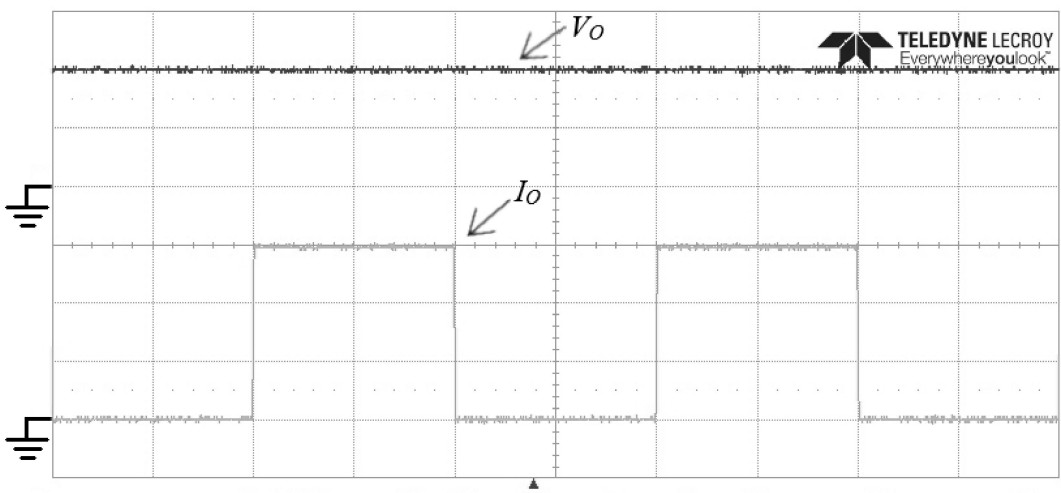

($V_o$: 5 V/div, $I_o$: 0.5 A/div, time: 50 ms/div)

**Figure 21.** Measured output voltage $V_o$ and output currents $I_o$ waveforms of the proposed hybrid converter operated in the discharging mode under step-load changes between 0% and 100% of full-load condition.

Comparison of conversion efficiency between flyback converter with hard-switching circuit and with the proposed active clamp circuit from light load to heavy load is shown in Figure 22, illustrating that the efficiency of the proposed converter is higher than that of hard-switching one. Its efficiency was 85% under full-load condition. According to component selection of the proposed hybrid converter, key component parameters are listed in Table 7. Power loss analysis of the proposed hybrid converter under full-load condition is illustrated in Table 8. Total power losses included switch, diode, transformer and driving circuit in the proposed hybrid one. The driving circuit loss was measured by oscillator and voltage $V_{cc}$ is 12 V and current $I_{cc}$ was 17.7 mA. The driving circuit loss $P_{DC}$ was 0.21 W. Since switches $M_1$ and $M_2$ were operated with ZVS at turn-on transition,

their switching loss only considered switching loss at turn-off transition. According to the maximum operational current and voltage of switch, diode and transformer in the proposed hybrid converter, losses of each component are listed in Table 8. Switch $S_1$ was one time in the turn-on or turn-off state during a day. Its loss was only conduction loss. According to (33) and Table 4, maximum flux density Bm can be determined and its value was 200 mT. Figure 23 shows the core loss curves of transformer $T_r$ manufactured by *PC95* material of TDK. When $B_m$ = 200 mT, core coefficient $C_p$ is equal to 110 mW/cm$^3$. Effective core volume $V_e$ of transformer $T_r$ was equal to 8.03 cm$^3$. The core loss could be determined and its value $P_{cTr}$ = 0.88 W. In addition, copper loss $P_{cpTr}$ could be obtained by (35). Since $I_{D1(\mathrm{rms})} = I_{S1(\mathrm{rms})}$ = 2.55 A, $I_{Lm2(\mathrm{rms})}$ = 6.5 A, $R_{dc1}l_{m1}$ = 0.027 Ω and $R_{dc2}l_{m2}$ = 0.062 Ω, copper loss $P_{cpTr}$ equaled 1.54 W. The conversion efficiency of the proposed hybrid converter operated in the discharging mode was 86.6% under full-load condition. The practical conversion was 85%. The stray loss of the proposed hybrid converter was 1.6%.

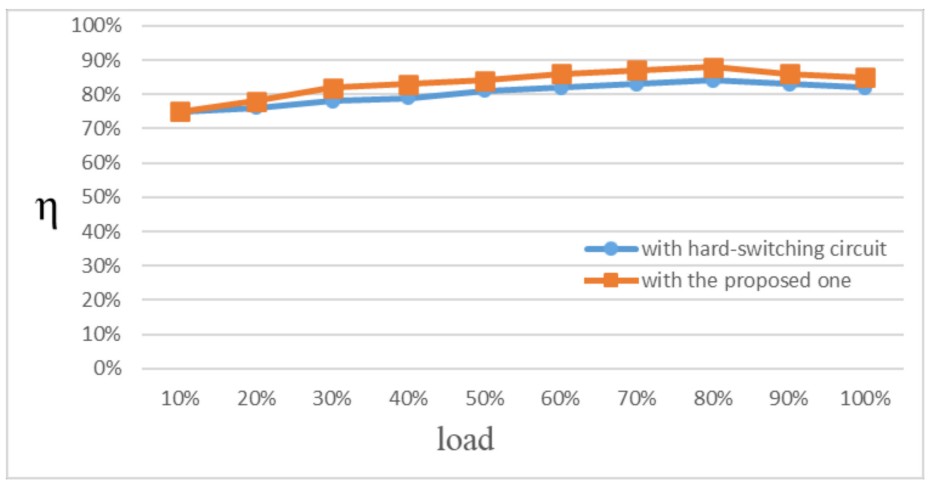

**Figure 22.** Comparison conversion efficiency between the conventional hard-switching flyback converter and the proposed one from light load to heavy load for operating in the discharging mode.

**Table 7.** Key component parameters of the proposed hybrid converter.

| Component | Part Number | Voltage/Current Ratings Or Formula | Features | | |
|---|---|---|---|---|---|
| | | | Symbol | Parameter | Values |
| $M_1, M_2$ $S_1$ | Aow2918 | 100 V/90 A | $R_{DS(on)}$ | Drain-source On resistance | <7 mΩ |
| | | | $t_{on}$ | Turn-on transition time | 41 nS |
| | | | $t_{off}$ | Turn-off transition time | 54 nS |
| $D_1$ | STPS10L60D | 60 V/10 A | $V_F$ | Forward drop voltage | 0.56 V |
| Transformer $T_r$ | TDK EE-33 (PC95 material) | $B_m = \frac{\mu_o \mu_r N I_{pk}}{(l_e + \mu_r l_g)}$ (T) $I_{pk} = \frac{N I_{o(max)}}{1 - D_{12(max)}} + \frac{\Delta I_{LM2}}{2}$ (A) $\mu_0: 4\pi \times 10^{-7}(\frac{H}{m})$ $I_{PK}$: maximum inductor current of primary winding (A) $I_{D(max)}$: output maximum current (A) $D_{12(max)}$: maximum duty cycle of switch $M_2$ $\Delta I_{Lm2}$:variation of current $I_{Lm2}$(A) $N$: turns of primary winding $A_e$: Effective core volume (m$^2$) | $\mu_r$ | permeability | 3300 |
| | | | $A_e$ | Effective cross-sectional area | 119 mm$^2$ |
| | | | $l_e$ | Effective magnetic Path length | 67.5 mm |
| | | | $V_e$ | Effective core volume | 8030 mm$^3$ |
| | | | $l_g$ | Air gap length | 0.8 mm |
| | | | $l_{n1}$ | Approximate mean length of turn in primary winding | 64 mm |
| | | | $l_{n2}$ | Approximate mean length of turn in secondary winding | 72 mm |
| | | | AWG#18 | Wire gauges of primary and secondary windings | Diameter 1.0 mm |
| | | | | $R_{dc}$ : resistance | 21.4 mΩ/m |

**Table 7.** *Cont.*

| Component | Part Number | Voltage/Current Ratings Or Formula | Features | | |
|---|---|---|---|---|---|
| | | | Symbol | Parameter | Values |
| | | | $N_1$ | turn of primary winding | 20 Turns |
| | | | $N_2$ | turn of secondary winding | 40 Turns |
| | | | $R_{dc1}l_{m1}$ | Resistance of primary winding | 0.027 Ω |
| | | | $R_{dc2}l_{m2}$ | Resistance of secondary winding | 0.062 Ω |

**Table 8.** Power loss analysis of the proposed hybrid converter under $V_B$ = 12 V and output maximum current $I_{o(max)}$ = 2 A.

| Component | The Proposed Hybrid Converter | | Total Power Losses of Each Component |
|---|---|---|---|
| | Symbol | Power Loss | |
| Driving circuit | $P_{DC}$ | $P_{DC} = I_{cc} \times V_{cc}$ $= 17.7 \times 12 \times 10^{-3} = 0.21$ w | $P_{DC} = 0.21$ w |
| Switch $M_1$ | Switching loss $P_{CM1}$ | $P_{CM1} = 0.15$ w | $P_{CDM1} = 0.24$ w |
| | Conduction loss $P_{DM1}$ | $P_{DM1} = 0.09$ w | |
| Switch $M_2$ | Switching loss $P_{CM2}$ | $P_{CM2} = 0.15$ w | $P_{CDM2} = 0.26$ w |
| | Conduction loss $P_{DM2}$ | $P_{DM2} = 0.11$ w | |
| Switch $S_1$ | Conduction loss $P_{DS1}$ | $P_{DS1} = 0.05$ w | $P_{DS1} = 0.05$ w |
| Diodes $D_1$ | Forward drop loss $P_{D1}$ | $P_{D1} = 1.43$ w | $P_{D1} = 1.43$ w |
| Transformer $T_r$ | Core loss $P_{cTr}$ | $P_{cTr} = 0.88$ w | $P_{cpTr} = 2.45$ w |
| | Copper loss $P_{cpTr}$ | $P_{cpTr} = 1.54$ w | |
| Efficiency | | 86.6% | |

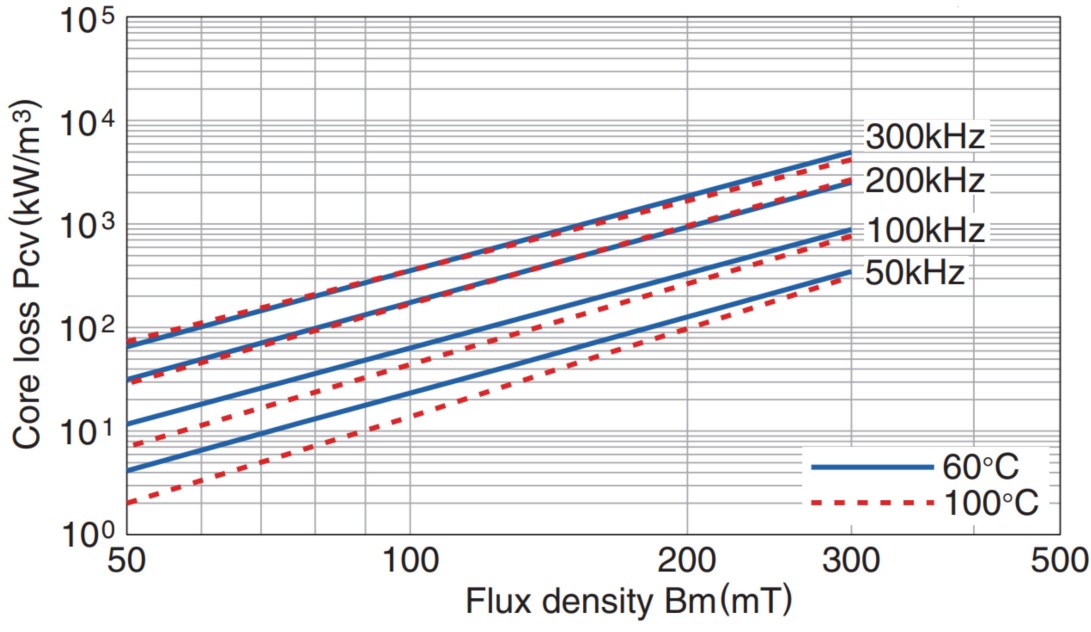

**Figure 23.** Core loss (mW/cm³) curves of transformer $T_r$ manufactured by *PC95* material of TDK.

## 6. Conclusions

The proposed hybrid converter is consisted of buck-boost converter and active flyback converter to implement battery charger and discharger. Circuit derivation of the proposed

hybrid converter is presented in this paper for decreasing component count. Moreover, operational principle, steady-state analysis, design and power loss analysis of the proposed hybrid converter have been described in detail. As compared with the conventional counterparts with hard-switching circuit, the proposed one can increase conversion efficiency of 4% and achieve efficiency of 85% under full load condition when the proposed one is operated in the discharging mode. In addition, cost and power density comparison between the proposed hybrid converter and the conventional counterpart converter, the proposed hybrid one can reduce cost of 19–22.7% and power density of the proposed one can increase about 1.5 times. An experimental prototype has been implemented for lithium battery of 12 V/3.2 Ah and for LED lighting of 10 V/2 A. It can verify the feasibility of the proposed hybrid converter. It is suitable for solar power applications.

**Author Contributions:** Conceptualization, methodology, and writing—original draft preparation, S.-Y.T.; writing—review and editing, J.-H.F. All authors have read and agreed to the published version of the manuscript.

**Funding:** This research was funded by Ministry of Science and Technology (MOST) in Taiwan grant number MOST 109-2622-E-182-003-cc3.

**Conflicts of Interest:** The authors declare no conflict of interest.

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
