# Peer review of "Buck-Boost/Flyback Hybrid Converter for Solar Power System Applications"

_electronics, doi:10.3390/electronics10040414_

Round 1

Reviewer 1 Report

The paper proposes a hybrid converter to supply power from solar power source to load. The paper is well written and well organized. However, the following are some questions and recommendations that the authors should address to improve the paper:
1- Please highlight a photo of your prototype converter in the paper.
2- The voltage and current stress of the proposed converter should be shown and compared with their counterparts.
3- Dose the proposed converter works in the grid-tied performance? If yes, please explain the operation and provide related waveforms.
4- A loss breakdown is necessary to evaluate the power loss of the converter. The loss breakdown should include the auxiliary power loss.
5- The authors should clarify the methodology and the equipment used to assess the converter efficiency.
6- Recently, the single-phase inverter idea has been shown helpful to fulfill the current and the voltage regulation. Some discussions on single-phase inverter should be provided, and these recently published results will provide some research ideas, e.g., High-buck in Buck and High-boost in Boost Dual-Mode Inverter, Single-Phase Dual-Mode Interleaved Multilevel Inverter for PV Applications, An Integrated Interleaved Dual-Mode Time-Sharing Inverter for Single Phase Grid Tied Applications.

Author Response

Dear Reviewer

Taking this opportunity, the authors would like to express our sincere appreciation to your suggestion regarding our manuscript entitled “Buck-Boost/Flyback Hybrid Converter for solar Power System Applications”. The revised manuscript has completely mostly according to suggestions and comments of reviewer. It is modified by red word to highlight. In the following, the modified contents of the revised manuscript are briefly described.

  1. According to the comments from the reviwers, the revised manuscript adds 4 tables.
  2. According to the comments from the reviwers, the revised manuscript adds 3 references and renews 7 references.
  3. According to the comments from the reviwers, the revised manuscript adds 3 figures.
  4. According to the comments from the reviwers, the revised manuscript deletes 1 figure, which Fig. 6(b) in the original manuscript.
  5. According to the comments from the reviwers, the revised manuscript added 17 equations for power loss analysis.

Finally, we point out from yours and try our best to explain some vague discussion which did not convince you in the attached file.

The form error has been modified in the revised manuscript.

Thanks again for your time and valuable comments.

With regards,

Sheng-Yu Tseng,

Associate Professor

Reviewer 2 Report

The manuscript sent for review on the buck-boost/flyback hybrid converter for solar power system applications is characterized by both high scientific level and contains a very large amount of knowledge mainly in terms of electronics, including related fields of electrical engineering and power electronics. 

The solution/technical approach proposed by the authors is based on the use of buck-boost/flyback hybrid converter for the purposes of supplying solar systems, both through in-depth analysis supported by a mathematical apparatus, conducted research and simulation tests, and confirmation of the obtained results, as well as analysis of the results achieved through a research experiment. 

The conducted research and simulation tests, on the basis of which the research results were obtained, were used by the authors for their detailed analysis and validation in order to implement the achieved solutions in practical applications. This is very important from the point of view of existing knowledge, as evidenced by the research results obtained by the authors in terms of the strategy adopted to solve the research problem. 

Both the research and simulation tests, as well as for the validation of the obtained research results, were conducted in the form of an experiment. The authors presented the research in two modes, namely for charging and discharging. In the next stage of the research, they performed a detailed analysis of the obtained results for both the simulation and the experiment, through the analysis of block diagrams and equivalents, as well as through the examination of the obtained waveforms of key parameters of the designed hybrid converter.

According to the recommendations of reputable publishers and journals, e.g. IEEE TTE, IEEE Access, Wiley and Sons, or MDPI, the abstract should have key elements, namely: introduction (reference to the research subject), unambiguous definition of the aim of the work, approximation/presentation of the potential solution of the problem/methods and reference on the basis of research findings (analysis, model, simulation/experiment) to the forecast of r obtaining research results in order to present relevant insights and formulations and final conclusions in terms of practical applications. 

Part of the abstract of an article, work, manuscript, etc. should not exceed 200 words, in this article there are 257 of them. In my opinion, this abstract does not refer to the observations and formulated conclusions (part of the end of the work), no clear definition of the aim of the work was formulated, and the methods contributing to solving the problem were not highlighted.

Comments on both parts of the abstract and the rest of the peer-reviewed work: 

  1. The abstract does not clearly state the goal of this study. 
  2. In the abstract of an article, paper, manuscript, abbreviations and markings should not be explained, e.g. ZVS.
  3. No unambiguity of the notation, e.g. in the abstract of this work 20W and 200 W, or 22.7 % and 85%. Please check the entire work in this respect.
  4. Duplication and explanation of abbreviations and designations, e.g. for the ZVS both in the abstract and in the keywords of this article. Please review the entire manuscript in this respect.
  5. Duplication of references to drawings, e.g. Fig. 5, p. 3, points 72-73, or Fig. 6(a) and 6(b) Please review all the work in this area and correct it if possible.
  6. The headings of the chapters, subchapters, e.g. chapter 5 should be moved from page 17, point 384, to page 18. I assume that the resulting situation was caused by the conversion to .pdf format. Please check all the work in this area. 
  7. Some of the abbreviations and markings cited in this manuscript are not explained, e.g. IEK, PWM or PWMIC in Fig. 14, etc. Please review the entire manuscript in this respect. 
  8. Please review this paper carefully for the use of punctuation marks and minor editing errors, e.g. p. 17, point 406 and p. 18, point 426, or p. 18, point 417 in case of 66ouH, etc., please also check the enclosed references.
  9. No unambiguity in description of drawings, e.g. Fig. 2 and Fig. 3 on page 2, or in case of Fig. 7 and Fig. 8 on page 5-6. Please check the whole article in this respect. 
  10. Incorrect description of drawings, e.g. Fig. 4 and Fig. 5 in the case of Schematic ... Please check the entire article for this.
  11. In the end of this work (conclusions) there is a lack of explicit and more detailed reference to the obtained results and their validation. This is all the more incomprehensible, as the article submitted for review is characterized by a significant scientific contribution, both the necessary simulation tests and the validation of the results obtained as a result of the research experiment were carried out in the scope of the research subject.

To sum up, the solution/technical approach proposed by the authors of this manuscript is supported by a thorough and detailed analysis of key parameters of the hybrid buck-boost/flyback converter together with a developed mathematical model and its schematics necessary for an in-depth analysis of the converter, supported by an accurate analysis of the physical phenomena occurring in the proposed converter through a proper graphical interpretation and tabulation of the obtained research results. 

The layout and structure of the manuscript are consistent, compact and properly developed in accordance with methodological and substantive principles. The great advantage of this paper is the presentation of the obtained research results both in a graphic form in case of block and conceptual diagrams (Figs. 1-7), the equivalent diagram and its waveforms of the proposed hybrid converter for charging (Figs. 8-9) and discharging (Figs. 10-11), as well as Figs. 12-13, presenting, according electrical circuits, diagrams, auxiliary circuits and simulation model of the designed converter in accordance with the research subject, and various waveforms of key parameters; as well as a tabulation of the results obtained from the research and tests carried out (Tables 1-3).

Strong aspects:

Technical approach; idea of solving the problem and its explanation; analysis of obtained research results, supported by mathematical analysis and formulation of final conclusions; well supported by analysis and experimental evidence; interesting for readers, stimulates new ideas; effective illustrations and tables; accuracy of applied methods and ability to use them.

Weak aspects:

Minor shortcomings that have no significant impact on the quality of the reviewed work, i.e. editing errors and poor quality of the methodological part of the abstract.

Recommended changes: 

Irrespective of the Editorial Board's decision, at this stage of the work, I would recommend that the authors of this paper correct any comments I have observed during the review of this manuscript contained above. 

Author Response

(The authors gave the same response as above.)

Reviewer 3 Report

In Line 31-32 it is stated that “Nowadays, due to a drastic increase in the demand for electricity, it leads to rapid and depletion of fossil fuels.“ In fact I would not agree that nowadays it is drastic increase in electricity demand. It is constantly seen for a number of year. The turn to the renewable energy sources is not because of depletion of fossil fuels. It is because of the absolutely other grounds.

Out of 15 references only 4 are within the last 5 year. Checking the topic in one of the publishers journals 100+ articles of the last 5 year (2016-2020) were found. Thus, I would recommend the authors to check more thoroughly the literature in order to see that is already achieved.

In additional 14 out of 15 are of the one Publishing House references, which is not the way it should be done.

I would suggest add in Fig. 15-19 axes. Also more explanation of those figures would be appreciated and put next to the figures.

Be aware of:

Line 9 – Since power generates for solar power.

Line 15 – two times proposed.

Line 42 – TFT is not explained.

Legends of some figures and tables starts not from the capital letter.

Line 307 - it can be seen – I would rephrase it.

Is logo of “Teledyne Lecroy” in figures necessary?

In Abstract and Section 5 it was indicated that power rating can be implemented till 200 W. However, seems I have missed evidence.

The real conclusion part is missing. In current version there is only statements what it is done.

Author Response

(The authors gave the same response as above.)

Round 2

Reviewer 1 Report

Authors have made changes in the manuscript as per reviewer comments. Now the paper qualifies for publication in MDPI Electronics.

I will suggest please go through the manuscript again to check minor grammatical and typo errors. Otherwise, the paper is now improved.

Congratulations!!

Reviewer 3 Report

Authors improved the manuscript considerably.

Still I would have wished that less percentage of references were from one Publishing House (in this particular case –  IEEE).

I wish very best to the authors in their research.